

# Advanced seismic characterization of a geothermal carbonate reservoir – Insight into the structure and diagenesis of a reservoir in the German Molasse Basin

Sonja H. Wadas[1], Johanna F. Bauer[1], Vladimir Shipilin[2], Michael Krumbholz[3], David C. Tanner[1], and Hermann Buness[1]

[1]Leibniz Institute for Applied Geophysics (LIAG), Stilleweg 2, D-30655 Hannover, Germany
[2]formerly LIAG; now at Geological Survey of North Rhine-Westphalia, De-Greiff-Straße 195, D-47803 Krefeld, Germany
[3]Independent researcher

**Correspondence:** Sonja H. Wadas (sonja.wadas@leibniz-liag.de)

**Abstract.** The quality of geothermal carbonate reservoirs is controlled by numerous factors and processes, such as the depositional environment, lithology, diagenesis, karstification, fracture networks, and tectonic deformation. Carbonatic rock formations are thus often extremely heterogeneous, and reservoir parameters and their spatial distribution are therefore difficult to predict. For the example of a 3D seismic dataset combined with well data from Munich, Germany, we demonstrate, how an

advanced analysis can deliver an improved reservoir model concept and help to identify possible exploitation targets within the Upper Jurassic carbonates. To identify possible reservoir sections and to understand their above-mentioned controlling factors, we analyse different seismic single- and multi-attributes. Some of the seismic attributes, together with lithology logs from wells, are then used to identify parameter correlations between the seismic attributes and the different carbonate lithologies to obtain a supervised neural network based 3D lithology model of the geothermal reservoir. Furthermore, we compare the

fracture orientations measured in seismic (ant-tracking analysis) and well scale (image log analysis), to address scalability. Our results show that, for example, acoustic impedance is well suited to identify reefs and karst-related dolines. Areas with strong karstification or fault- and fracture-related deformation, which are both associated with high permeabilities, are also indicated by e.g. strong frequency attenuation, variance anomalies, and/or morphological features like bowl-shaped structures derived from the shape index. Furthermore, by using sweetness we can reconstruct the reef development of two exemplary reefs, and

regarding the lithology distribution, we show that the upper part of the reservoir is dominated by limestone and dolomitic limestone rather than dolomite. In addition, we observe spatial trends in the degree of dolomitization. With respect to the fracture orientations on seismic- and well scales, we point out that a general scalability is not possible due to a combination of methodological limitations and geological reasons. Nonetheless, we argue that the combination of both methods provides an improved overall impression of the fracture systems, and therefore possible fluid pathways. By taking all the results into account, we are

able to improve and adapt recent reservoir concepts to outline the different phases of its structural and diagenetic evolution. Furthermore, our results help to identify high quality reservoir zones in the Munich area.



## 1 Introduction

The quality of a geothermal carbonate reservoir is controlled by different factors and processes, such as the depositional environment, lithology, diagenesis, karstification, fracture networks, and tectonic deformation (Andres, 1985; Lemcke, 1988;
Mraz, 2019). Carbonate rock formations are thus often extremely heterogeneous (Birner et al., 2012; Konrad et al., 2019; Bohnsack et al., 2020; Fadel et al., 2022) and important reservoir parameters, such as reservoir volume, porosity, permeability, temperature, and their spatial distribution, are difficult to predict (Veeken, 2007; Huenges, 2010; Agemar et al., 2014; Glassley, 2014; Moeck, 2014; Bauer et al., 2019). However, a good understanding of these parameters and their distribution are required for a successful geothermal project (Backers et al., 2022; Fadel et al., 2022).

Most carbonate reservoirs are located within deposits of a former shallow marine environment, e.g. the Upper Jurassic carbonates of the German Molasse Basin. Shallow marine carbonates can be often separated into two hyper facies types, a massive facies consisting of reefs and a bedded facies consisting of layered carbonates (Reinhold, 1998). Another distinguishing feature is the lithology type. The main process that alters the lithology type in carbonates after deposition is dolomitization (Machel, 2004; Lucia, 2007). During this process, calcium is replaced by magnesium. Depending on the degree of dolomitization, car-
bonate rocks can be assigned to different lithology types: limestone (90% to 100% $CaCO_3$), dolomitic limestone (50% to 90% $CaCO_3$), calcareous dolomite (50% to 90% $CaMg(CO_3)_2$), and dolomite (90% to 100% $CaMg(CO_3)_2$). Dolomitization can lead to a reduction of the rock volume and therefore, to an increase of the total porosity by creating secondary porosity (Sajed & Glover, 2020). Furthermore, early dolomitization can increase rock strength and thus preserve primary porosity by creating a stable framework which hinders compaction (Lucia, 2007). Therefore, dolomitization can cause heterogeneity of petrophysical
rock properties (Ehrenberg & Nadeau, 2005; Ehrenberg, 2006) and a redistribution of the pore space (Mountjoy & Marquez, 1997). Dolomite is also more resistent against erosion and less soluble than limestone, which makes it less prone to karstification (Steidtmann, 1911). The influence of dolomitization on the porosity distribution has been investigated by many studies. For example, for the Jurassic carbonates in southern Germany Bohnsack et al. (2020) and Wadas & von Hartmann (2022) have shown that, in order of decreasing porosity, the dolomitic limestone has the highest porosities, followed by limestone, and
dolomite and calcareous dolomite have the lowest porosities within the carbonate reservoir.

Another reservoir-quality control factor is karstification, which describes the dissolution of calcite or aragonite due to the percolation of unsaturated meteoric- or groundwater, e.g., caused by fluid migration along fracture zones, or a falling sea level and resulting subaerial exposure of the carbonates. This process can improve the reservoir quality by enlargement of primary pore space and thus fluid pathways, and the formation of secondary porosity and large cavities, which can also develop into
dolines (Kendall & Schlager, 1981; Xu et al., 2017). Karstification is often more intense close to faults because of the often increased fracture intensity (Closson & Abou Karaki, 2009; Del Prete et al., 2010; Wadas et al., 2017).

Permeability is also strongly controlled by fractures, which are often associated with tectonic- and fault-related deformation. However, permeability provided by fractures can also lead to unwanted fluid flow behaviour like a channelized fluid flow, which can cause an early thermal breakthrough (e.g., Toublanc et al., 2005; Jolley et al., 2007; Bauer et al., 2019; Boersma





et al., 2020; Fadel et al., 2022). Therefore, understanding the local and regional fracture network, e.g. the fracture orientations, connectivity, and the fracture density, is of high importance when characterizing complex reservoirs (Boersma et al., 2020).

Normally, reservoir parameters are derived from well data or outcrop analogues (Bauer et al., 2017). Wells can deliver direct information of the local reservoir properties such as porosity, permeability, fracture orientation and -intensity, lithology, and facies types, but sparsely-located wells are unable to depict the spatial distribution of the properties. A method that is espe-

cially suited to depict the spatial changes related to geological and petrophysical variations is 3D seismic attribute analysis of 3D reflection seismic (Chopra & Marfurt, 2007; Ashraf et al., 2019; Zahmatkesh et al., 2021). Seismic attributes are quantities derived from seismic data, based on e.g., time, amplitude, frequency, phase, velocity, and attenuation (Chopra & Marfurt, 2007). For a long time they have been used for hydrocarbon reservoir characterization, prediction and monitoring, by delivering information on geological structures, lithology, reservoir properties, parameter relationships and patterns that might not be rec-

ognized otherwise (Taner, 2001; Chopra & Marfurt, 2007; Sarhan, 2017). For example, Banerjee & Ahmed Salim (2020) used seismic attributes to analyse the structural features and the depositional patterns of the NW Sabah Carbonate Platform in the South China Sea. Using spectral decomposition, they identified paleo-lows and depocentres; sweetness helped them to identify channels, reef structures and lithofacies boundaries; variance and amplitude extraction maps revealed the reefal development on top of the carbonate platform. Al-Maghlouth et al. (2017) used frequency decomposition and a colour-blend of geometric

attributes, such as semblance and conformance, to characterize the Cenozoic carbonates facies in North-West Australia, and to define edges and discontinuities associated with depositional geometries, such as reefs. Spectral decomposition and coherency have also been used by Skirius et al. (1999) to locate faults and fractures, and reef margins and isolated buildups, as targets for increased hydrocarbon production, e.g. at the Leduc carbonate reef bank in Canada and the Tor Field area in the North Sea. Furthermore, Wang et al. (2016) used seismic attributes to describe the reef growth and the evolution of channel systems

for Eocene carbonates in the Sirte Basin in Libya. Edge- or discontinuity detection attributes, like variance and chaos, have been used to conduct a small-scale seismic-based fracture analysis (Jaglan et al., 2015; Williams et al., 2017; Albesher et al., 2020; Boersma et al., 2020; Loza Espejel et al., 2020). Nonetheless, seismic data lacks the high vertical resolution of well and outcrop data. Various studies have therefore attempted to show the benefits of a combined approach using both seismic and well data, in order to reduce the uncertainties of reservoir characterization (Toublanc et al., 2005; Fang et al., 2017; Albesher

et al., 2020; Boersma et al., 2020; Méndez et al., 2020). Another aspect to consider is that manual interpretation of seismic data can be a very time-consuming task due to the high amount of data, which is why computational solutions, such as supervised and unsupervised neural networks have been increasingly used for seismic interpretation, pattern recognition, and lithology classification in recent years (Saggaf et al., 2003; Baaske et al., 2007; Bagheri & Riahi, 2015; Roden et al., 2015; Brcković et al., 2017; Zahmatkesh et al., 2021). Besides the long-time use for hydrocarbon reservoir investigation, seismic attribute

analysis has also been increasingly used in geothermal exploration in recent years, especially for complex structured reservoirs (Pendrel, 2001; Chopra & Marfurt, 2007; Doyen, 2007; Abdel-Fattah et al., 2020), e.g., in Poland (Pussak et al., 2014) and Denmark (Bredesen et al., 2020). Another such complex carbonate reservoir is located in the South German Molasse Basin.

Based on a case study in Munich, Germany, we perform an advanced analysis of a 3D seismic dataset and well logs for the Upper Jurassic geothermal carbonate reservoir, which has beeen described as a complex lithology- and facies-dependent,



fracture-and karst-controlled reservoir by other studies (Birner et al., 2012; Cacace et al., 2013; Homuth et al., 2015; Moeck et al., 2020). The aim of this study is to deliver an improved reservoir model concept and to identify possible exploitation targets within the Upper Jurassic carbonates. To accomplish this goal, this paper is structured in three main parts. First, several seismic single- and multi-attributes are analysed to identify and better understand the physical and structural reservoir characteristics. Second, parameter correlations between the seismic attributes and the different carbonate lithologies are investigated to obtain

a supervised neural network based 3D lithology model of the geothermal reservoir. And third, a seismic fracture orientation analysis (FOA) workflow is adapted, based on other studies (e.g. Albesher et al., 2020; Boersma et al., 2020), and applied to the 3D seismic dataset. The FOA results on seismic scale are compared with those at image log scale, in order to address the scalability of the FOA results.

## 2  Study site

The study site of the GRAME 3D seismic dataset covers about 170 km$^2$ and is located below the city of Munich within the South German Molasse Basin (Fig. 1a). This area includes the geothermal plant Schäftlarnstraße (Sls) that consists of six horizontally-deviated wells (three injection and three production wells, Fig. 1b). The reservoir section of the Upper Jurassic carbonates (Malm) is at a depth of around 1750 to 2600 m depth below sealevel (1380 to 1750 ms two-way traveltime (TWT)).

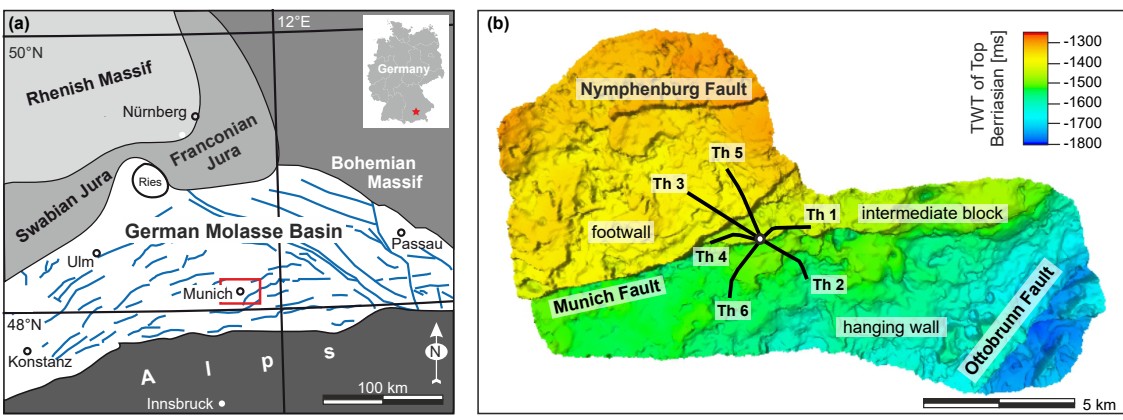

**Figure 1.** Geological context and study area. (a) Simplified overview of the German Molasse Basin with tectono-stratigraphic units and major faults (blue lines). The study area and the location of the GRAME 3D seismic are marked by a red star/rectangle. (b) Further input data is gathered from geophysical logging of the geothermal site 'Schäftlarnstraße (Sls)' in the city of Munich, which contains six horizontally deviated wells (Th1 to Th6). Based on the horizon interpretation of the 3D seismic data carried out by Ziesch (2019), the Upper Jurassic reservoir is located at a depth between 1380 to 1750 ms TWT or around 1750 to 2600 m depth below sealevel.



### 2.1 Geological evolution

The German Molasse Basin (GMB) is part of the North Alpine Foreland Basin, which experienced a complex structural evolution with a Permo-Carboniferous graben phase, a Triassic to Middle Jurassic epicontinental or shelf phase, a Middle Jurassic to Cretaceous passive margin phase, and a Tertiary foreland phase (Bachmann et al., 1987; Bachmann & Müller, 1992). During the Upper Jurassic, up to 600 m of carbonates, partly forming reef buildups, were deposited in the study area, which at that time was covered by the Tethys Sea (Schmid et al., 2005; Pieńkowski, 2008). At the Jurassic-Cretaceous boundary, regression of the Tethys led to the exposure of the carbonate deposits and caused strong karstification (Bachmann et al., 1987; Bachmann & Müller, 1992). Several isolated sinkholes and also sinkhole clusters can be observed at fault terminations or within the reservoir (Lemcke, 1988; Ziesch, 2019; Wadas & von Hartmann, 2022). From Late Cretaceous to Early Tertiary, southern Germany was uplifted, resulting in extensive compressional deformation due to the Alpine Orogeny. As a result, antithetic and synthetic normal faults developed parallel to the Alpine front (Ziegler, 1987). Furthermore, the underthrusting of the European plate below the Adriatic-African plate in the Late Eocene caused the Alpine nappes to extend to the north, which led to isostatic-induced downflexing of the GMB (Frisch, 1979). The development of the GMB was accompanied by two major transgressive-regressive cycles (Eisbacher, 1974), each causing the accumulation of marine deposits followed by terrestrial sediments e.g. from rivers and lakes, known as 'Marine Molasse' and 'Freshwater Molasse', respectively. Finally, the Molasse sediments were overlain by Pleistocene glacial and interglacial deposits.

### 2.2 Geothermal reservoir

The southward-dipping Upper Jurassic carbonates (Malm) form a carbonate platform which is the geothermal reservoir (Schmid et al., 2005; Pieńkowski, 2008). Due to the southwards increase in depth, the temperatures increase towards the south from approximately 70°C to 150°C. This allows extraction of geothermal energy for heating in the Munich area and even electricity generation further south (Böhm, 2012; Agemar et al., 2014; Homuth et al., 2015).

With regards to the depositional system, the Malm of the greater Munich area can be separated into two hyper facies: a massive facies (consisting of reefs) and a bedded facies (Reinhold, 1998; Machel, 2004; Lucia, 2007; Bohnsack et al., 2020). Several studies have shown that reefs are suitable exploitation targets because they are often prone to dolomitization, karstification, and brittle deformation, and therefore often exhibit enhanced groundwater flow (Andres, 1985; Stier & Prestel, 1991; Birner et al., 2012). A lithofacies-based, hydrostratigraphic classification for the greater Munich area carried out by Böhm (2012), shows that the lowermost units Malm $\alpha$ to Malm $\gamma$, which mainly consist of marly limestones, can be characterized as an aquitard. Malm $\delta$ to Malm $\epsilon$ are described as a regional aquifer due to a laterally-persistent dolomitic massive facies. Malm $\zeta$ contains local aquifers in dolomitized massive facies as well as aquitards.

Structural analysis of the GRAME 3D seismic dataset reveals that the study area is influenced by karstification and fault-related deformation. Several isolated sinkholes and also sinkhole clusters can be observed along the faults and at fault terminations (Sell et al., 2019; Ziesch, 2019; Wadas & von Hartmann, 2022). The seismic data also shows that the greater Munich area is traversed by a complex fault pattern, mainly consisting of normal faults. The largest fault with a maximum offset of 350 m



is the Munich Fault (Fig. 1b), which splits eastward into several branches that subdivide the area around the geothermal site 'Sls' into a footwall block, an intermediate block and a hanging-wall block. The large fault system in the southeast is the Ottobrunn Fault that also splits into several small antithetic and synthetic faults with small offsets up to 80 m forming a horsetail

splay. This indicates this fault has both normal fault- and strike-slip components (Ziesch, 2019). Overall, the Upper Jurassic carbonates of the Munich area are characterized as a strongly heterogeneous, lithology- and facies-dependent, fracture- and karst-controlled reservoir (Birner et al., 2012; Cacace et al., 2013; Homuth et al., 2015; Moeck et al., 2020).

## 3    Methods

For a better insight into the structure and diagenesis of the reservoir we performed an advanced seismic data analyses of the

GRAME dataset. The GRAME 3D seismic dataset, which was surveyed and processed by DMT Petrologic GmbH (Scholze & Wolf, 2016a, b), had a variable line distances of 400 to 500 m, and a source and receiver spacing of 50 m with a sweep frequency of 12 to 95 Hz, 5 seconds of record length and a 2 ms sample rate. This configuration enabled the acquisition of a high-resolution 3D cube (for details, see tabular overview of acquisition parameters and processing steps in Wadas & von Hartmann (2022)). The advanced seismic data analyses performed in this study comprised the following techniques: single-attribute

analysis, multi-attribute analysis, attribute and neural-network based lithology classification, and ant-track based seismic fracture orientation analysis. See also the appendix for more detailed information regarding the seismic attributes and the chosen parameters for the attribute analysis.

### 3.1    Seismic single-attribute analysis

Seismic attributes are quantities computed from the seismic data and describe the shape or physical characteristics of one

or more seismic traces, mostly over specified time intervals. The characteristics of a seismic wave, e.g. velocity, amplitude, frequency, phase, and attenuation, change while the wave propagates through the subsurface. These changes are caused by the different physical rock properties of the various rocks in the subsurface. Therefore, they are used to highlight specific geological, physical, and/or reservoir properties, and to help recognize patterns and parameter relationships (Taner, 2001; Chopra & Marfurt, 2007; Veeken, 2007). Seismic single-attributes can be categorized by different taxonomies (Dewett et al.,

2021). In this work, we group them by the properties they measure, namely amplitude-related attributes, phase- and frequency-related attributes, and discontinuity-related attributes.

**Amplitude-related attributes:** These attributes (e.g., Root mean Square (RMS) amplitude, envelope, reflection intensity, and acoustic impedance) are used to depict stratigraphic and lithologic contrasts. The RMS amplitude is described as the square root of the sum of squared amplitudes divided by the number of samples within a specified time window. The reflection

intensity is the average amplitude over a specified time window multiplied with the sample interval. And the envelope is the magnitude of the complex trace and is independent of phase (Chopra & Marfurt, 2007; Sarhan, 2017). These three attributes mainly identify only strong anomalies (see appendix figure A1) and to depict smaller variations, as expected in our study area, we also used the acoustic impedance. Every reflection changes the amplitude of the returning wave due to a contrast in



acoustic impedance, which is the product of the seismic velocity of the wave travelling through the subsurface and the density of the rock. Therefore, the reflection amplitudes can be inverted to get impedance values (Pendrel, 2001; Barclay et al., 2008; Filippova et al., 2011) by using, e.g., a stochastic seismic amplitude inversion, as carried out in this study area by Wadas & von Hartmann (2022).

**Phase- and frequency-related attributes:** Phase- and frequency-related attributes (e.g., instantaneous phase, instantaneous frequency, and dominant frequency) show the continuity of weak reflectors and indicate unconformities, faults, fracture zones, lithology- and stratigraphic sequences, and sequence boundaries (Van Tuyl et al., 2018). The instantaneous phase measures the phase shift of a specific reflection event, e.g. resulting from a polarity reversal of the reflection coefficient or due to a curved interface (Chopra & Marfurt, 2007). The instantaneous frequency is the rate of change of the instantaneous phase, and the dominant frequency is the square of the instantaneous frequency summed with the square of the instantaneous bandwidth and then the square root of the sum is calculated (Schlumberger, 2020). So the dominant frequency indicates where the energy of the seismic signal is concentrated in the frequency domain.

**Discontinuity-related attributes:** Such attributes (e.g. variance and chaos) are able to image vertical and lateral discontinuities, and can therefore highlight stratigraphic and structural boundaries like faults, fractures, dolines, and reef edges (Chopra & Marfurt, 2007). For the variance, a trace-by-trace analysis is performed in order to quantify the dissimilarity of the seismic waveform of neighboring traces within a specified time window (Bahorich & Farmer, 1995; Marfurt et al., 1998; Wang et al., 2016). Chaos is a similar attribute that maps the 'chaoticness´ of the seismic signal from statistical analysis of dip/azimuth estimates.

## 3.2 Seismic multi-attribute analysis

The complexity of carbonates makes a combined analyses of several different attributes necessary in order to characterize and better understand the reservoir. In a multi-attribute analysis, several single-attributes, which are mathematically independent but linked through the underlying geology, can be combined by corendering and colour-blending, or by using them to calculate new attributes (Marfurt, 2015). Since a good seismic attribute should represent important aspects of the underlying geology, showing more than one attribute in the same image can give improved geological insight. See also the appendix for more detailed information regarding the seismic attributes and the chosen parameters for the attribute analysis.

**Sweetness:** Sweetness can enhance the visibility of lithological and structural changes (Chen & Sidney, 1997). It is the mathematical combination of instantaneous frequency and envelope (Radovich & Oliveros, 1998), and the combination of these two attributes is able to detect general energy changes of the seismic wave. High sweetness values are correlated with both high envelope and low instantaneous frequencies, whereas for low sweetness values it is the opposite.

**Spectral decomposition:** Spectral decomposition is a useful tool for qualitative and quantitative interpretation, because it allows to delineate geologic features in more detail due to tuning at a specific frequency (Henderson et al., 2008; Wang et al., 2016). Therefore, it can enhance subtle structural features like thin beds, reefs, channels, and pinch-outs (Chopra & Marfurt, 2007; Marfurt & Kirlin, 2001; Marfurt, 2015), and it can also be used for seismic geomorphology analysis (Marfurt & Kirlin, 2001). In principle, spectral decomposition (Fig. A2) separates the seismic signal into its frequency components. Three



different frequencies are then corendered and displayed by RGB colour-blending, where frequencies 1, 2 and 3 are plotted against red, green and blue, respectively (Chopra & Marfurt, 2007; Al-Maghlouth et al., 2017).

**Curvature:** Curvature is used to detect changes in depositional- and geological trends, e.g. tectonic features and lineaments. Therefore it is useful to identify, e.g., channels/valleys, karst/dolines, and mounds/reef buildups (Al-Dossary & Marfurt, 2006). Curvature measures how much a seismic reflector is bent. In case of a planar reflector, the corresponding vectors are parallel and the reflector has zero curvature. In contrast, anticlinal features result in diverging vectors and the curvature is defined as positive. Synclinal features result in converging vectors and the curvature is defined as negative (Roberts, 2001; Al-Dossary &

Marfurt, 2006; Wang et al., 2016). Most positive ($K_{pos}$) and most negative ($K_{neg}$) curvature are often corendered and used for morphological analyses (Chopra & Marfurt, 2007; Marfurt, 2015).

**Shape index & Curvedness:** Additionally, Curvature can be used to calculate new attributes such as the shape index and the curvedness that allow a quantitative definition of the local morphology of a seismic reflector/seismic horizon independent of scale (Roberts, 2001; Chopra & Marfurt, 2007; Veeken, 2007; Wang et al., 2016; Sarhan, 2017). The Shape Index describes

the type of shape of a specific surface and the index differentiates between dome (+1), ridge (+0.5), saddle (0), valley (-0.5) and bowl (-1) shapes (Chopra & Marfurt, 2007; Marfurt, 2015). The curvedness measures the magnitude of curvature which is zero for a plane surface (Roberts, 2001)). Both attributes are often corendered together with variance or chaos to visualize reflector morphology (Marfurt, 2015).

### 3.3 Neural network based lithology classification

The lithology classification in this study is based on a rock-physics workflow, which predicts the most probable lithologies using a supervised neural network. 3D seismic volumes, well logs and geological interpretations can serve as input datasets. At first, a class definition (class 1: limestone, class 2: dolomitic limestone, class 3: dolomite, class 4: calcareous dolomite) using well logs is carried out, after which the classification algorithm can be generated, and finally used for lithology prediction (Schlumberger, 2020).

Before applying the rock physics-based class definition, it needs to be tested if parameter relationships between the lithology classes and the physical parameters derived from the seismic attributes exist. This was accomplished by cross-plotting the lithology logs from the six wells against the seismic attributes that were extracted from the 3D volumes along the well paths. A correlation analysis was carried out and the correlation coefficients between the lithology classes and the attributes were determined. Based on the results, the following attributes were chosen: acoustic impedance, dominant frequency, reflection in-

tensity, variance, envelope, and the 28 Hz frequency band, and then a cluster analysis was carried out to develop a classification algorithm. This cluster analysis was performed by utilizing a supervised artificial neural net, which automatically searched for the best relationships between the seismic attribute values and the lithology classes. Such a supervised neural network is trained by providing both the input data (seismic attribute 'logs') and the desired output data (lithology logs). From the neural-network based cluster analysis, 2D and 3D probability density estimates were derived (Figs. 2a & b).

Training the estimation model was an iterative process and at the end of each iteration the training error was calculated. The error was assessed by comparing the neural network result with the given target, in this case the lithology logs. At the



beginning, the training data was split into two parts, one part was used for the training and the other was used to calculate the error by cross validation (Schlumberger, 2020). The chosen ratio was 50:50, so 50% of the input data were used for training of the neural network and the other 50% were used to cross validate the results. To quality control the performance of the

classification algorithm a confusion matrix was generated (Fig. 2c). It allows the user to identify whether the algorithm gets confused when defining the different classes (Sammut & Webb, 2017). The confusion matrix (prediction/true table) shows the probability of a classification occurring given the true class. The rows contain the true classes and the columns contain the predicted classes. For example, 8.18% of the samples that belong to the Limestone class were wrongly predicted as a Dolomite, and 86.17% of the samples that belong to the Dolomite class were correctly predicted as a Dolomite. Whereas, the

true/prediction table of the confusion matrix shows the data the other way around with the probability of a sample belonging to a particular class given the predicted class. High-confidence classification results will have large values along the diagonal of the matrix, as it is the case in our study, indicating a reliable classification result. Furthermore, the data quality was also visually inspected by comparing the actual lithology logs from the six wells with estimated logs based on the neural network and it also shows a good classification result (Fig. 2d). The successfully trained neural net was then used to create a 3D model

with the predicted lithologies based on the 3D seismic attribute volumes (Schlumberger, 2020).

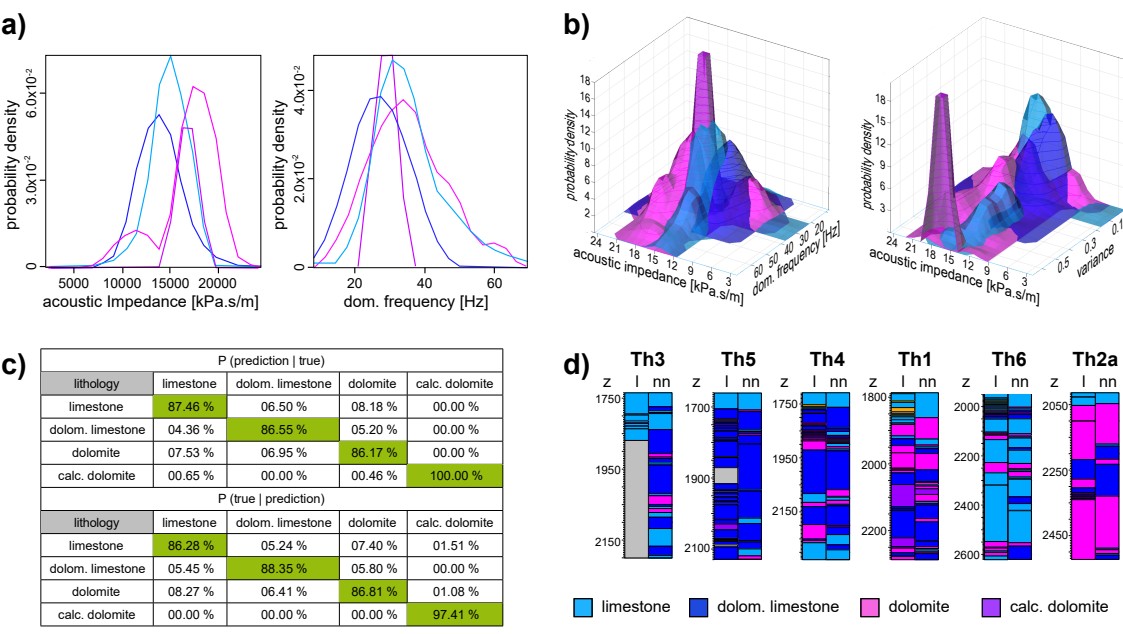

**Figure 2.** A lithology classification using a supervised neural network is based on parameter relationships between the input data (e.g., seismic attributes) and the desired output data (e.g., lithology classes from logs). (a & b) After choosing appropriate seismic attributes for the classification, these attributes were used to create probability density estimates, e.g. in 2D and 3D (exemplarily shown for acoustic impedance, variance, and dominant (dom.) frequency). The results of the lithology classification were validated by examining (c) a confusion matrix table and (d) a comparison between the actual lithology log (l) and the predicted lithology log (nn) derived from the neural net.



## 3.4 Ant-track based fracture orientation analysis

Several studies have shown the applicability of 3D seismic data for fracture analysis (Jaglan et al., 2015; Fang et al., 2017; Albesher et al., 2020; Boersma et al., 2020; Loza Espejel et al., 2020; Méndez et al., 2020), which can be used to address the often-raised question of scalability of reservoir properties (e.g. Lake & Srinivasan, 2004; Li et al., 2019), i.e., can information

regarding fracture properties be transferred from the well scale to the seismic scale and vice versa? To address this question, an adapted seismic fracture orientation analysis (FOA) workflow based on e.g. Albesher et al. (2020) and Boersma et al. (2020) was applied. Afterwards the FOA results on the seismic scale were compared with those on the image log scale.

The FOA workflow, based on the 3D reflection seismic volume, consisted of several steps, starting with the extraction of an edge-detection volume, in this case, variance (Fig. 3 – step 1), which images lateral and vertical discontinuities (Marfurt et al.,

1998; Chopra & Marfurt, 2007). Then the edge evidence attribute was applied to enhance the edges and thus, the fractures in the variance volume (Fig. 3 – step 2). The edge evidence attribute searches for segments in which the variance values differ significantly from the surrounding values, in order to enhance them (Schlumberger, 2020). Afterwards Schlumberger's patented ant-tracking algorithm (Schlumberger, 2020) was applied (Fig. 3 – step 3). Ant tracking can be used to extract faults and fractures from a discontinuity volume. This is accomplished by simulating the behaviour of ants, which use pheromones to

optimize their search for food by marking their paths. Following the same principle, artificial ants (agents) are used to search for faults and fractures generating a detailed attribute volume with very sharp edges. A choice can be made between a passive and an aggressive ant tracking mode, and since our investigation focus is on small structures like fractures and not large structures such as faults, we chose an aggressive mode and adapted its parameters to our data (see additional explanation in appendix). After the 1st ant tracking a 2nd ant tracking was applied to further sharpen the detected edges (Fig. 3 – step 4). Afterwards

automatic fracture extraction (Schlumberger, 2020) was used to create 3D fracture patches from the ant-tracked volume (Fig. 3 – step 5). Then the fractures from the patch volume were extracted along the well paths including an area with a diameter of one kilometre around each well, in order to capture enough fractures for the analysis (Fig. 3 – step 6). Afterwards, visual quality control was carried out and fracture patches that deviated from the ant tracks were manually removed from the dataset (Fig. 3 – step 7). The fracture patches along the well paths (Fig. 3 – step 8) and their corresponding fracture orientation values were then

exported from Petrel and imported into the software Stereo32 to create rose diagrams and stereogram pole plots in order to analyse the fracture orientations and to determine fracture clusters. Subsequently, the results were compared for the complete drill paths and sections of it, with those of image log analyses, to show where the fractures matched. The Compact Micro Imager (CMI) images the rock's electrical resistivity and is presented as an unrolled figure of the wells surface (Schlumberger, 2004). Since fractures have a different resistivity compared to the surrounding rock, they are visible as sinusoids. We traced

their orientation manually by using the software WellCAD (ALT, 2021). The image is referenced according to the well path, so that the software calculates the true orientation of the fractures.





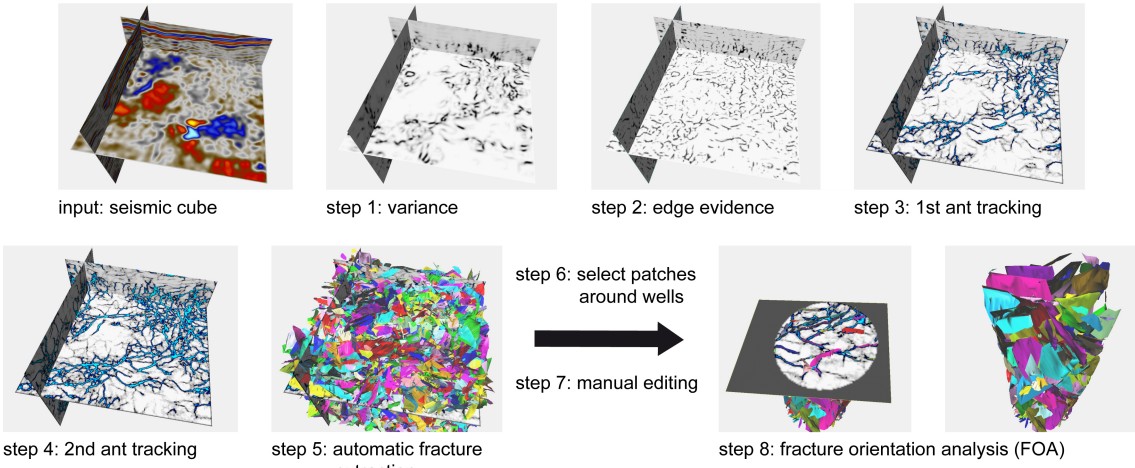

**Figure 3.** Fracture orientation analysis workflow, shown on a zoomed-in section of the GRAME dataset, based on seismic attributes and ant-tracking.

## 4 Results

In the following, the results of the physical and structural reservoir characterization, the lithology prediction model, and the fracture orientation comparison are shown. Please note that the carbonate formations of the GMB generally dip to the south and additionally, our study area shows a fault-related downward-stepping to the south. Therefore, the carbonate deposits on the footwall of the Munich Fault are located at shallower depths/time slices compared to the hanging wall.

### 4.1 Physical and structural reservoir characterization

#### 4.1.1 Geological main targets

In the GRAME area, several of the main exploration targets can be identified based on the acoustic impedance volume, e.g., reef buildups and dolines, as described by Wadas & von Hartmann (2022). For example, in the west of the footwall block, an elongated reef (Fig. 4a) with high impedance values of up to 18000 kPa*s/m in the reef core and low impedance values down to around 10000 kPa*s/m at the reef margins can be seen. Comparable characteristics are observable for another reef in the east of the intermediate block (Fig. 4b). The identified dolines are typically characterized by a circular shape. In the study area, large dolines and doline clusters are often located close to faults. This is because the fracture zones around faults can lead to enhanced fluid migration, and therefore increased dissolution of soluble rocks and the development of secondary porosity, as it can be observed, e.g., at the Munich Fault (Fig. 4c, d) and the Ottobrunn Fault (Fig. 4e). The dolines have low impedance values in the center and often larger impedance values around them. Another observed feature is an incised narrow channel in the upper reservoir part in the east of the hanging wall, crossing the area from northeast to southwest (Fig. 4f). This channel is associated with a low-stand sea level and subaerial exposure of carbonates. The channel fill is clearly visible



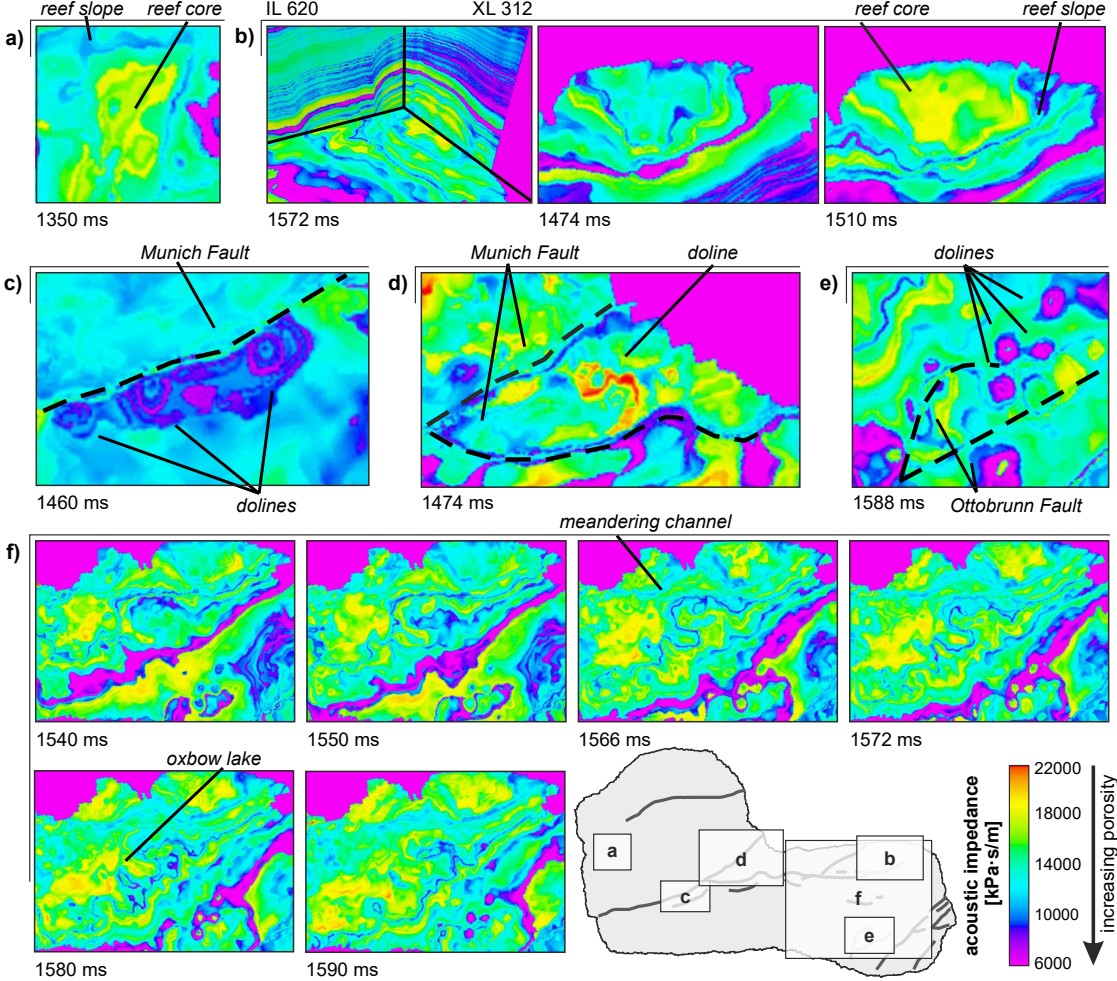

**Figure 4.** Time slices through the acoustic impedance volume, calculated by Wadas & von Hartmann (2022), which is used to depict stratigraphic and lithological contrasts. The results reveal impedance contrasts of different carbonate structures associated with a shallow marine environment and sea-level variations, such as reef buildups (a, b), karst features (c, d, e) and a channel (f). Low impedance values correlate with high porosities. IL = inline and XL = crossline.

due to its low impedance values, which show a strong contrast compared to the impedance of the surrounding rocks. In the deeper time slices, e.g., at 1590 ms the channel is only slightly curved and follows a more or less straight northeast-southwest direction. Over time, the channel started to meander (e.g. at 1580 ms and 1572 ms), also resulting in the development of oxbow lakes. A derived porosity model based on an impedance/porosity relationship which associates low impedance values with higher porosities shows a complex porosity distribution within the study area (Wadas & von Hartmann, 2022). The reef core

has mainly porosities < 3% and the highest porosities of 7 to 14% are observed at the reef cap, in the upper third of the reef and on the reef slopes. Wadas & von Hartmann (2022) assume that this is the result of intense karstification and gravitational





mass flows on the slopes. Overall, the footwall of the Munich Fault shows higher porosities than the hanging wall to the south, and the porosity also displays a W-E trend with higher porosities in the western part of the study area. Furthermore, based on a comparison of their results with well data, they describe that dolomitic limestone has the highest porosities and calcareous

dolomite has the lowest porosities. Thus, preferential exploitation targets in terms of high porosity are more likely to be found in the upper part of the reservoir (Berriasian to Malm $\zeta$1), in particular, in dolomitic limestones due to their high porosity and/or in strongly karstified areas within bedded and reef facies.

Besides amplitude/impedance, other physical properties describing the signal of a reflected seismic wave are phase and frequency. In our study area noticeable changes of seismic phase are observed at interface boundaries of dolines and reefs.

Additional to the already identified dolines and doline clusters (Fig. 4), we found smaller circular features in the north of the footwall which were also interpreted as dolines (Fig. 5a). The already identified dolines at the Munich Fault (Fig. 5b) and at the Ottobrunn Fault (Fig. 5c) also produce clear phase changes of the seismic signal. Furthermore, two additional circular features can be seen north of the already identified doline cluster at the Ottobrunn Fault. Phase changes are also observed at reef boundaries and within the reefs. For example, the reef to the east of the intermediate block (Fig. 5d) shows a clear phase

change at the outer reef edge, but internally, further phase changes can be observed, in part almost parallel to the outer edge. Whereas in the center, there is an area with a laterally almost constant phase, e.g., at 1470 ms. This is the reef core and the surrounding phase changes reflect the shifting of the reef edge, and thus the reef growth that has led to the enlargement of the reef over time. Furthermore, at the southeastern edge of this reef, the areas of the same phase have sometimes a significantly smaller lateral extent than at the southwestern edge, e.g., at 1490 ms and 1500 ms. This indicates that in the southwest of the

reef the same lithology/facies occupies a larger spatial area than in the southeast. Thus, we assume that this reef has either not uniformly spatially grown or was affected by spatially uneven erosion, due to e.g. spatial differences in water motion, sediment dynamics, and/or subaerial exposure.

With regard to the investigation of the dominant frequencies in our study area, it is shown that the frequencies are mostly below 40 Hz, indicating strong frequency attenuation. Nevertheless, the degree of frequency attenuation differs, even for the

same type of structural feature. For example, a reef buildup in the west of the hanging wall (Fig. 6a) shows much lower dominant frequencies of mostly less than 20 Hz, compared to the reef in the east of the intermediate block (Fig. 6b) with frequencies of mostly around 20 to 40 Hz. For the latter, only the reef slopes and the cap have frequencies below 20 Hz. This correlates with the results of the impedance analysis, which show that these zones are characterized by lower impedance values, and therefore higher porosities (Figs. 4a & b). This might result from mass redistribution at the slopes and more

intense karstification leading to stronger attenuation of higher frequencies. For the reef in the west of the hanging wall, this means that it might be more karstified. The correlation of karstification and low frequencies is also observable for the large dolines at the Munich Fault (Fig. 6c) and the Ottobrunn Fault (Fig. 6d). The doline margins are characterized by very low frequencies of mostly below 20 Hz, while the doline center shows slightly higher frequencies of around 25 to 30 Hz. As shown by other doline studies, their margins are often strongly fractured (e.g. Waltham et al., 2005; Al-Halbouni et al., 2018; Wadas

et al., 2018) leading to a loss of high frequencies. Besides reefs and karst, the filling of the narrow channel, identified in the





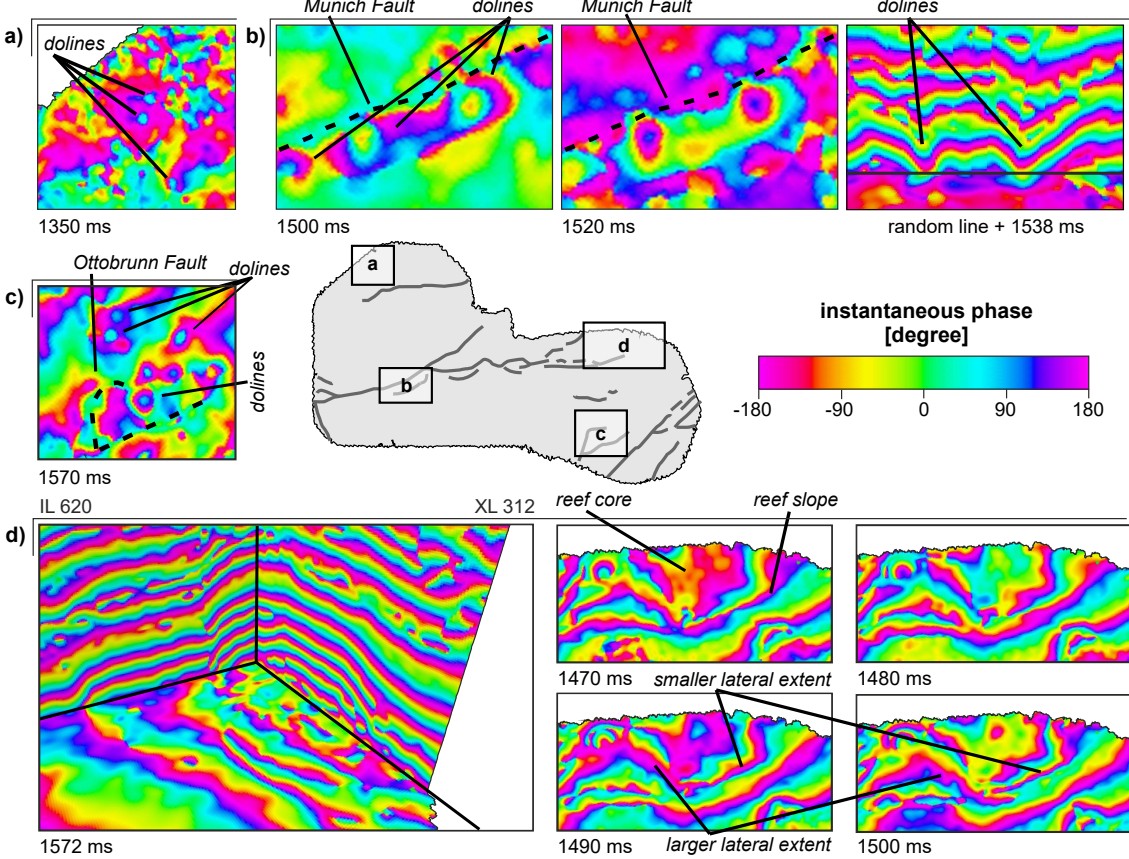

**Figure 5.** Time slices through the instantaneous phase volume, which is used to identify phase shifts in the data. Strong phase changes, independent of amplitude, are observable at structural and interface boundaries, e.g. dolines (a, b, c) and reefs (d). IL = inline and XL = crossline.

impedance volume, is also characterized by lower frequencies compared to the surrounding material (Fig. 6e). Although the frequencies differ along the channel indicating variations of the channel fill.

Another way to analyse the frequency content of a seismic dataset is by spectral decomposition, because the examination of individual frequency components enables a better spatial differentiation and correlation. The reef buildups, the karst features

(dolines and widespread dissolution along bedding planes) and also the meandering channel are all characterized by mainly low frequencies of 18 Hz and partly high frequencies of 50 Hz represented by red, pink and to a lesser degree by purple colours (Fig. 7). This frequency attenuation due to scattering at e.g., fractures and reef edges, could be caused by intensive karstification and mass redistribution. In addition, lithological variations can also lead to frequency differences. Regarding regional trends, the footwall of the Munich Fault (Fig. 7a) shows a north–south differentiation towards the deeper time slices. The northern part

of the footwall is mainly dominated by the 28 Hz frequency component and partly by the 50 Hz frequency component, which is represented by the green and green-blue colours. The southern part of the footwall is characterized by increasingly lower





**Figure 6.** Time slices through the dominant frequency volume showing the varying frequency attenuation across the study area, e.g. (a & b) within reefs, (c & d) dolines, and (e) channels. Lower dominant frequencies indicate increased karstification and/or fault-related deformation. IL = inline and XL = crossline.

frequencies towards deeper time slices, more exactly at 1358 ms all three frequency components are present, at 1380 ms the 28 Hz component dominates together with the 18 Hz component, and at 1396 ms the 18 Hz band dominates. The upper part of the hanging wall of the Munich Fault (Fig. 7b) shows a west–east frequency differentiation. The western part contains mostly the 18 Hz and partly the 50 Hz frequency components, and only in greater depths the 28 Hz component is also present. To the east, a bright coloured area can be seen, e.g., at 1524 and 1544 ms, indicating that all three frequency components are present and no general loss of high and/or low frequencies occurs. Overall, the hanging wall of the Munich Fault is more dominated



by the blue 50 Hz frequency band than the footwall, the only exception for the hanging wall is the easternmost area containing the Ottobrunn Fault, which shows lower frequencies.

**Figure 7.** Time slices through the spectral decomposition volume with 18 Hz plotted against red, 28 Hz plotted against green, and 50 Hz plotted against blue. It enables the examination of individual frequency components, showing spatial frequency variations (a) on the footwall and (b) on the hanging wall of the Munich Fault.



### 4.1.2 Internal reef architecture

To investigate the internal reef architecture the seismic multi-attribute sweetness was analysed. In our study area the bedded facies is characterised by low sweetness values (brown, dark blue and grey colours), indicating a low envelope and high frequencies (Fig. 8). In contrast, the reefs that show a strong variation in sweetness, are characterized by a higher sweetness due to increased envelope values compared to the surrounding material. The reef base (Fig. 8a – 1462 ms) and parts of the reef core show very high sweetness values (white, light blue and yellow colours). This results from very high envelope values

combined with relatively low frequencies. Such a combination of envelop and frequency could indicate a mudstone or a generally compacted and cemented (fine-grained, microcrystalline) limestone (Dunham, 1962; Flügel, 2010). The reef core has medium to high sweetness values resulting from medium envelope values and low to medium frequency values (Fig. 8; light green, pink, orange, medium blue and red colours). This indicates a mixed mud- and grain-supported carbonate texture which is typical for a reef core that consists of different types of biogenic components like sponges, bivalves, corals and

bryozoans surrounded by a matrix (Flügel, 2010; Böhm, 2012; Homuth et al., 2015). The different biogenic components cause variations in rock properties, such as rock density or seismic wave velocity, which result in stronger amplitude differences and lower frequencies due to increased attenuation. The reef slopes have medium sweetness values (dark green and purple colours), resulting from low frequency and low to medium envelope values. This corresponds to other studies which have shown that reef slopes often consist of a grain-dominated and strongly-disturbed debris facies (Flügel, 2010; Playton et al., 2010), which

will lead to low internal reflection coefficients and strong frequency attenuation. The sweetness attribute also allows a more detailed interpretation of the reef development. For example, in the case of the reef to the west of the hanging wall (Fig. 8a), it is observed that, starting from the reef base at 1462 ms, two closely spaced but separate reef cores developed (1452 ms and 1448 ms), ultimately forming one large reef (1440 ms). The western reef core shows a bending orientation where the southern part has a SW–NE orientation and the northern part has SE–NW orientation. Whereas, the eastern reef core shows only a

NW–SE orientation. In contrast, the reef to the east of the intermediate block (Fig. 8b) shows a slightly different development. The reef base, which is not shown, is spatially coherent and also the reef core (from 1490 ms to 1460 ms) had a spatially mostly uniform development. However, in the upper part, at 1428 ms and 1416 ms, the reef core starts to develop the shape of a three-armed star with NNW–SSE, N–S, and NE–SW orientations. During further reef growth the NE–SW arm died out and only a NW–SE oriented reef core remains (1400 ms). This change in reef core shape might result from a change in ocean currents or

other local environmental changes.

### 4.1.3 Structural boundaries & lineaments

Variance and chaos are able to highlight structural boundaries and lineaments. However, both attributes deliver comparable results, so only the results of the variance analysis are shown. High variance values indicate high dissimilarities between neighbouring traces, especially on the footwall (Fig. 9a) and the intermediate block of the Munich Fault (Fig. 9b). Long linear

features and chaotic patterns are visible along the fault itself and in the surrounding areas, respectively. The former results





**Figure 8.** Time slices through the sweetness volume, which detects general energy changes of the seismic wave in order to identify changes in lithology and stratigraphy. For the studied reservoir, the internal architecture of two exemplary reefs (a) in the west of the hanging wall and (b) to the east of the intermediate block of the Munich fault are shown, revealing a complex reef development. IL = inline and XL = crossline.

from the strong discontinuities due to the fault displacement, and the latter is induced by fault-related deformation. Fault-related deformation can cause an increase in fracture intensity which generate more discontinuities, which is observable by high variance values. But discontinuities can also be caused by karst-related deformation, as it can be seen, e.g., for the dolines at the Ottobrunn Fault (Fig. 9c) and the Munich Fault (Fig. 9d). In the top view, the strongly disturbed doline edge is shown by



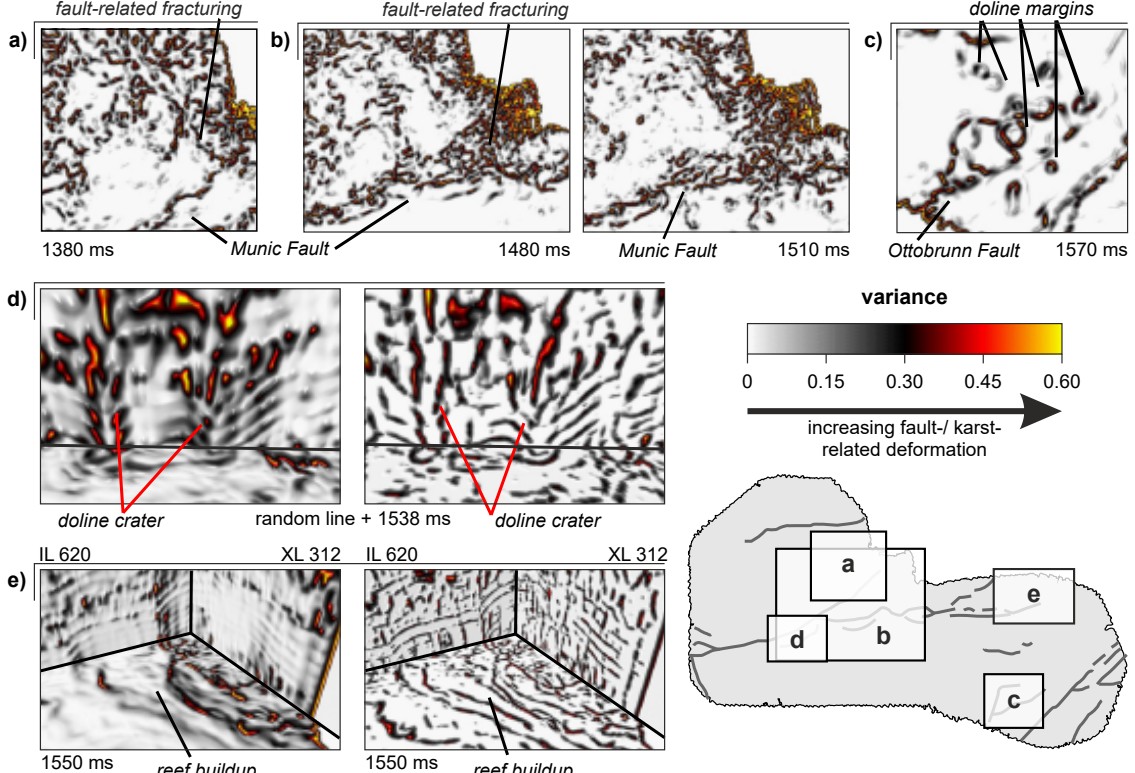

**Figure 9.** Time slices through the variance volume, which is used to detect faults and fracture zones (a & b), karst features (c & d) or other structural features with sharp edges, such as reefs (e). In (d) and (e) the second subfigures on the right show variance with an additionally-applied edge-enhancement filter to better highlight the vertical discontinuities on an inline (IL), a crossline (XL) and a random line.

circular high variance values and in the side view, the collapse crater is clearly visible due to vertical discontinuities. In addition, reef margins are also characterized by high variance values (Fig. 9e). These are presumably caused by mass redistribution due to debris flows at the margins, intensified karstification, and fracturing.

### 4.1.4 Morphological features

To analyse morphological features within the reservoir, we used most positive curvature (Fig. 10a) and most negative curvature (Fig. 10b) corendered with variance (Fig. 10c). In our study area, most positive curvature reveals many anticlinal and most negative curvature reveals many synclinal features, as shown by an exemplary time slice at 1510 ms. Along the Munich Fault and it's fault branches, long linear features can be seen (Fig. 10c). The positive curvature anomalies are associated with the upthrown side of the normal faults and the negative curvatures with the downthrown side (Roberts, 2001). And the distance between both anomalies gives an impression of the fault heave. Since the distance is quite small, this indicates a short lateral displacement of the fault, which also fits well with the finding that the faults in this study area are steeply dipping (Ziesch,





2019). Besides this, many small-scale lineaments that form a chaotic pattern are visible, especially on the intermediate block and the footwall. These small linear features are interpreted as fractures resulting from intense fault-related deformation along the Munich Fault, in particular within the area where the fault splits into two fault branches forming the intermediate block. Along this fault, but also along the Ottobrunn Fault to the south, and within the fault blocks, many circular features can be seen, which are interpreted as karst-related dolines (Figs. 10d & e). The doline margins show a distinct positive anomaly indicating an anticlinal structure, whereas the doline center shows a negative anomaly which is typical for a synclinal structure.

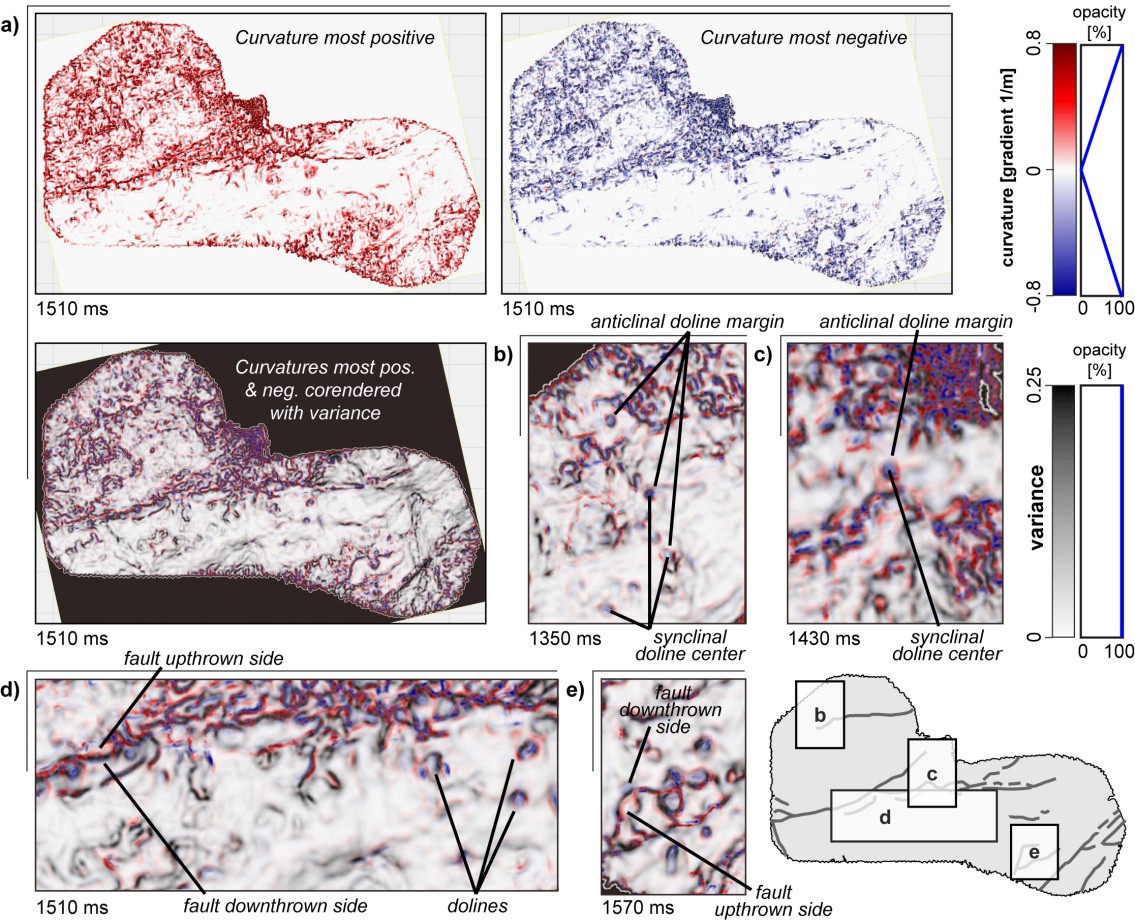

**Figure 10.** Time slices through the curvature volumes that are utilized to detect changes in structural - and depositional trends. (a) The most positive curvature reveals anticlinal and (b) the most negative curvature synclinal features. (c) For a comprehensive interpretation both curvature volumes are superimposed and corendered with variance. The joint interpretation allows the detection of faults and the determination of the upthrown and downthrown sides, and it is also a suitable tool for doline detection (d – e).

Curvature distinguishes only between planar, anticlinal and synclinal structures, but it can be used to calculate new attributes, such as the shape index and the curvedness (Fig. 11) that allow a quantitative definition of the local morphology (Roberts, 2001). Therefore, we corendered the shape index with curvedness and variance. In our study area, zones with planar or almost





planar features, shown by dark grey colours, are observable on the hanging wall of the Munich Fault. Zones with highly-curved morphologies, visible by bright colours, are seen on the footwall and the intermediate block of the Munich Fault, and to the south at the Ottobrunn Fault, e.g. at 1510 ms (Fig. 11a). The tectonically-deformed fractured area in the east of the footwall

**Figure 11.** Time slices through the co-rendered shape index and curvedness volumes (left panels), also together with the variance volume (right panels). The shape index describes the type of shape of a specific surface and the curvedness measures the magnitude of curvature. Together with variance they are suited to visualize morphology (Marfurt, 2015), e.g. like (b) damage zones, (c & e) karst-related dolines, (d & e) and faults.

block is characterized by a chaotic pattern of mainly ridge- and valley-shaped surfaces (Fig. 11b), but we also identified many small-scale bowl-shaped structures that are distributed across the entire area. Larger bowl-shaped dolines are only located in the





north of the footwall. They have a ridge-shaped outer margin and a valley-shaped inner margin that changes to a bowl-shaped
structure in the centre (Fig. 11c). Along the Munich Fault itself the footwall is characterized by a ridge-shaped lineament
and directly adjacent to the south is a valley-shaped lineament, which defines the transition towards the intermediate block.
Although this transition zone is less pronounced regarding morphology than the transition zone between the intermediate block
and the hanging wall (Fig. 11d). To the south at the Ottobrunn Fault, the shape index shows that the identified horse-tail splay

(Ziesch, 2019) has a valley-shape in the west and a ridge-shape in the east, showing that the fault is downthrown to the west
(Fig. 11e). The identified large dolines show the same characteristics regarding their shape as the large dolines to the north.
The shape index also shows other small bowl-shaped objects, some of which can also be found along the fault and due to
the lack of a valley-shaped margin they have only a low morphological contrast compared to the surrounding area. They are
either comparatively small dolines with a shallow collapse crater or small and shallow sagging structures, which do not have a

collapse crater and therefore no strong curvature change at the margin.

### 4.2   Lithology prediction model

The main process influencing the lithology type in carbonates is dolomitization (Machel, 2004; Lucia, 2007). In the study area
four types of carbonate with regards to their degree of dolomitization can be found: limestone, dolomitic limestone, dolomite,
and calcareous dolomite. For the study area, the 3D lithology model derived from the neural network classification reveals

that calcareous rocks (limestone and dolomitic limestone) with a total fraction of 76% are more common than dolomitic rocks
(dolomite and calcareous dolomite) with a fraction of 24% . The examination of vertical cross-sections (Fig. 12a – c) shows
considerable variations in lithology distributions, especially within the reef buildups, e.g. on the intermediate block and the
footwall of the Munich Fault. These show that reef cores that are only slightly porous consist of mainly dolomite. However,
also interbedded sequences with limestone and dolomitic limestone occur, especially in the upper part of the reef buildups, but

sometimes also in areas closer to the reef base.

By investigating individual time slices in the top view (Fig. 12d & e), it is also noticeable that areas can be defined in
which certain lithology types are dominant. For the entire reservoir of the GRAME study site, the upper part is dominated by
limestone and dolomitic limestone, both for the footwall block (e.g. at 1380 ms) and for the hanging-wall block (e.g. at 1540 ms
and 1580 ms). Apart from that, it is noticeable that the footwall block shows higher degrees of dolomitization in the west and

northwest compared to the east and southeast (e.g. at 1460 ms). On the other hand, the hanging wall shows the opposite
trend with increased dolomitization to the east and in the central part (e.g. at 1600 ms and 1620 ms), although this spatial
trend diminishes towards greater depths (e.g. at 1660 ms). Overall, dolomitization appears to be slightly more pronounced
on the hanging-wall block than on the footwall block. Additionally, the subdivision of the reservoir into a lower part with
almost completely dolomitized carbonates and an upper part with more partially dolomitized carbonates, could indicate several

dolomitization and dedolomitization phases.

With regard to the reefs, the time slices show the same lithology distribution as observed in the cross sections, with mainly
dolomite in the reef cores and dolomitic limestone and limestone at the reef slopes and in the upper parts of the reefs (e.g. at
1380 ms and 1580 ms). According to the results of Wadas & von Hartmann (2022), who showed that the reef slopes and the



**Figure 12.** Cross-sections and time slices through the 3D lithology model derived from the neural network classification of the seismic attributes. The model enables the interpretation of the vertical and lateral distribution of the different carbonate types and allows the identification of dolomitization trends at local scale within reefs (a – c) and at regional scale on the footwall (d) and on the hanging wall of the Munich Fault (e).





reef caps have higher porosities compared to the reef cores due to karstification, it is assumed that limestone and dolomitic

limestone in the reefs appear to be more prone to karstification than the areas of pure dolomite. Similar observations can be made in areas that comprise mostly of limestone-dominated carbonates and show intense karstification, e.g., the sinkhole cluster in the north of the footwall block (e.g. at 1380 ms) or the western part of the hanging wall (e.g. at 1540 ms) with karstification along bedding planes according to Wadas & von Hartmann (2022).

In addition, we could not find any other correlations between structural characteristics and the spatial distribution of dolomi-

tization. For example, the amount of dolomite is not increased at faults, or intensely fractured areas with increased permeability, such as in the east of the footwall block. This indicates that dolomitization in the Munich area is more facies-controlled than fault-controlled.

### 4.3  Comparison of FOA derived from well and seismic data

An important question that is often raised in reservoir exploration addresses the scalability of fracture properties like the

orientation and size. To address this question, we compared fracture orientations (FO) in the vicinity of the well paths derived from the seismic attribute analysis with the CMI results. Results are in the following grouped according to the tectonic blocks, to facilitate comparisons within the three fault blocks of the Munich Fault.

On the footwall the fracture orientations (dip direction/dip angle) and derived fracture clusters for the wells show both good and poor agreement. Regarding Th3 only the seismic results are shown due to data restrictions of the corresponding image

interpretations. Overall, the seismic data (Fig. 13) reveals three fracture clusters at 346/01, 290/01, and 233/01. However, we observe a change in the fracture systems with depth. In the depth range between 1730 m and 1810 m, we observe two fracture cluster at 338/01 and 67/06. In the depth range between 1810 m and 2050 m, we observe three fracture clusters, the main cluster (173/11) is comparable to the upper part, but we observe two new clusters at 72/09 and 310/26. In the deepest part between 2050 m and 2190 m, we observe a broad range in fracture orientations and no distinct clusters can be identified. For

the well Th5, which was drilled mainly in dolomitic limestone, the fracture clusters show a variable match between the seismic and image analyses. (Fig. 13). In general, the seismic data shows with four clusters a more diverse distribution in fracture orientations than the CMI data with just two clusters. A good match with ±15° is accomplished for the 282/01 cluster (from seismic data) and the 284/03 cluster (from CMI data). A moderate match with ±30° is observed for the 159/05 fracture cluster (from seismic data) and the 176/06 cluster (from CMI data). Two other fracture orientation clusters that we observe in the

seismic data are not present in the CMI data. To analyse wether the variable match holds for the complete well path or not, we also compare fracture orientations within different depth sections separately. We observe that in the upper depth range between 1680 m and 2000 m that consist of bedded and massive facies, one of the two clusters shows a good match. A second cluster identified in the seismic is not observable in the well. In the lower part from 2000 m to 2130 m, which is mostly of bedded facies, the fracture clusters show a poorer match, whereby in the seismic the E-W striking fracture set is split into two

(conjugate) sets.

On the intermediate block, a strong well-dependent distinction with respect to matching fracture clusters between seismic- and CMI data is evident. For well Th4, which was drilled mostly in dolomitic limestone of the massive facies and to a minor



degree of limestone and dolomite, a poor cluster match is found with the closest match for one cluster within an angle of
±30° (Fig. 13). Notably, fracture orientations are stable along the entire well path. In contrast, Th1 (Fig. 13) in the east of the
intermediate block, which was drilled in dolomite, dolomitic limestone and calcareous dolomite, shows a good fit of fracture
orientations, with two clusters fitting with ±15° (seismic data: 280/06 & 214/05; CMI data: 273/06 & 44/04) and one matching
with ±30° (seismic data: 337/03; CMI data: 139/03). Similar to Th5, the quality of the cluster match between CMI and seismic
data changes with depth, i.e. the fit between the clusters is better in the upper part of the reservoir (1840 m to 2180 m), compared
to the lower part (2180 m to 2290 m). The upper part consists mostly of a bedded and a mixed facies, whereas the lower part
consist of massive dolomite.

On the hanging-wall block (Th2a and Th6), when comparing fracture orientations over the entire well paths, it is observable
that the quality of the match between seismic and CMI data is better for the well Th6. The Th6 was drilled mostly in massive
limestone and in the upper part to a minor amount in bedded limestone. In the seismic data we found three fracture clusters that
show either a good (seismic: 240/04; CMI: 233/01) to moderate (seismic: 117/02; CMI: 137/01) fit with ±15° to ±30°, or no
fit (002/16) with the CMI-data (Fig. 14). Furthermore, we observe variations of the fit with respect to depth. For example, the
upper and lower part of Th6 have both clusters that match good within ±15°, and clusters that match moderate within ±30°.
In the middle part the fit is poor, i.e. only two clusters match within ±30°. For the entire well Th2a (Fig. 14), drilled mostly
in massive dolomite and massive dolomitic limestone, only a poor cluster fit is found. Only one fracture cluster fits within
±30° (seismic: 333/12; CMI: 001/04). Additionally, in the seismic as well as in the CMI data one fracture set exists, that was
not identified by the other method. The depth dependent fracture fit shows that in the upper part between 2020 m and 2400 m
depth at least the rough orientation fits. In the lower part a reliable comparison is not possible because of the low number of
measurements in the CMI data. However, both methods show a roughly NE–SW and a NW–SE striking fracture set.

In summary, correlations between lithology/facies and well location are evident with respect to the matching of fracture
clusters. Fracture orientations in massive limestone show a better match than in massive dolomite. Independent of facies, the
degree of dolomitization seems to correlate negatively with scalability. With regard to the well locations, differences can be
detected within the fault blocks in terms of corresponding fracture orientations. On the footwall, Th3 in the west is located
in mostly undisturbed material compared to Th5 which is situated in disturbed rocks, according to the discontinuity-attribute
analysis. Furthermore, the area around Th5 also has slightly higher porosities than the area around Th3 according to the porosity
model (Wadas & von Hartmann, 2022), indicating that the area around Th5 is more affected by fault- and probably karst-related
deformation. This led to the generation of fractures with more diverse orientations, as observed for Th5. Therefore, Th5 shows
a poorer match than other wells. Based ont this assumption, we expect a much better FO match for Th3 because the well
and its surrounding are situated in mostly undeformed material and as a result, the preferred fracture orientations should also
show up more clearly in the seismic data compared to Th5. On the intermediate block, Th4 shows almost no fitting fracture
orientations compared to Th1, which is probably also a result of more intense fault-related deformation, because Th4 is located
at the branching point of the Munich Fault where it splits into two fault branches. On the hanging wall, Th6 in the west shows
a slightly better FO match than Th2a probably due to slightly enhanced fault-related fracturing more to the eastern part of
the fault block. Overall, the seismic analysis is able to distinguish several fracture orientation sets/groups, similar to the CMI



**Figure 13.** Comparison of fracture orientations from seismic and CMI data from the wells on the footwall (Th3 and Th5) and the intermediate block (Th4 and Th1) of the Munich Fault using rose diagrams and stereographic pole plots. Seismic fracture orientations are shown by white rose-diagrams and CMI fracture orientation are shown by grey rose-diagrams, together with colour-coded fracture cluster matching. For each well the lithologies, the facies types, and the stratigraphy are given.





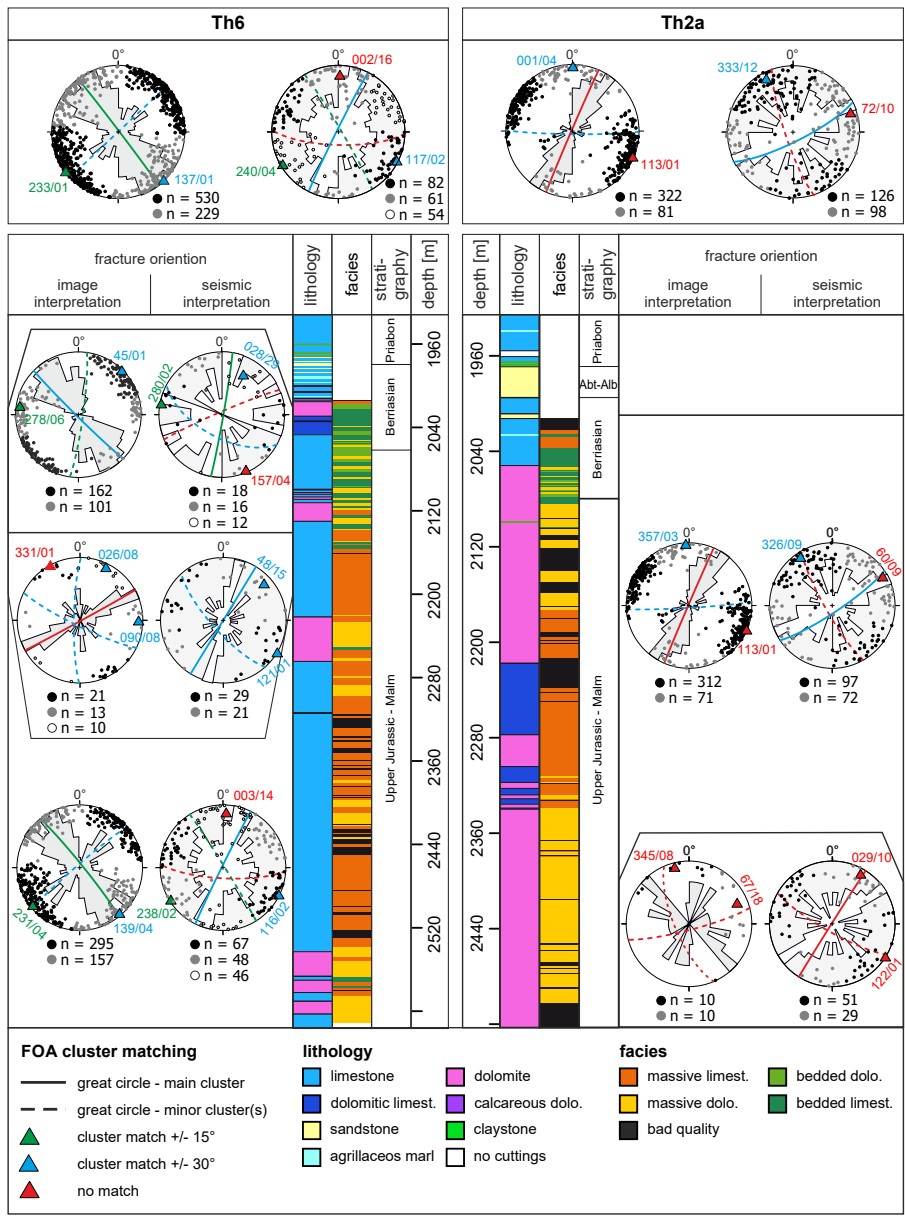

**Figure 14.** Comparison of fracture orientations from seismic and CMI data for the wells on the hanging wall (Th2a and Th6) of the Munich Fault using rose diagrams and stereographic pole plots. Seismic fracture orientations are shown by white rose-diagrams and CMI fracture orientation are shown by grey rose-diagrams, together with colour-coded fracture cluster matching. For each well, the lithologies, the facies types, and the stratigraphy are given.

analysis, namely a NNE–SSW oriented fracture set, a NE–SW oriented fracture set, an ENE–WSW oriented fracture set, a NW–SE oriented fracture set and a NNW–SSE oriented fracture set.



## 5   Discussion

The benefit of seismic attributes is to highlight contrasts in the data. However, the more complex a reservoir is, the more difficult it is to interpret these contrasts. Due to their strong heterogeneity, the characterization of carbonate reservoirs is therefore a major challenge in exploration (e.g. Ehrenberg & Nadeau, 2005; Lucia, 2007). In the following, first the chosen methodical approach is inspected and afterwards we discuss the scalability of fracture orientations, and the structural and diagenetic evolution of the reservoir. Taking all results into account, we provide new exploitation targets in the Munich area for possible future geothermal projects.

### 5.1   Methodical approach

We have demonstrated the benefits and a number of applications of seismic attribute analysis for the characterization of a geothermal carbonate reservoir in the GMB. In the following we discuss aspects regarding the calculation and the usage of seismic attributes that need to be considered.

Seismic attributes are a quantitative measure of the seismic data. They are sensitive to geology, and thus allow to draw conclusions about the structural interpretation or characterization of the depositional environment, e.g., faults, stratigraphy and geomorphology (Chopra & Marfurt, 2007). Since their first application in the 1970s, a large number of seismic attributes have been developed, which makes it difficult to select the appropriate ones for a specific analysis (Chopra & Marfurt, 2007). According to Barnes (2006), the following criteria should be considered to avoid the use of unnecessary attributes: discard duplicate seismic attributes and if there are multiple attributes that measure the same property, choose the one that works best for your data, and prefer attributes with geological or geophysical meaning. To avoid duplicate attributes in our study, crossplot analyses were not only performed between the attributes and the lithology logs, but also between the attributes themselves. In case of a linear or quadratic relationship, which indicates that the attributes contain nearly the same information, one of the attributes was discarded from further analyses. Furthermore, all chosen attributes have geological/geophysical meaning linked to the reservoir characterization.

The effectiveness of seismic attribute analysis is also controlled by the type of reservoir. The great advantage of seismic attributes is to highlight contrasts in the data. However, the more complex a reservoir is, e.g., regarding control factors (reef development, karstification, and dolomitization), the more difficult it is to interpret these contrasts. Therefore, prior knowledge of possible reservoir control factors that may affect the physical properties of the seismic signal is important when performing the seismic attribute analysis itself. Almost all attribute analyses can be adapted to the object under investigation to obtain viable results by changing parameters such as the inline and crossline radii, or the number of samples or traces (Chopra & Marfurt, 2007; Schlumberger, 2020). If the goal is, e.g., the interpretation of large-scale faults, rather large spatial parameters should be chosen, e.g., in the variance analysis a lateral range of 8 inlines and crosslines should be used. Whereas for the investigation of fractures, the analysis should be carried out on a smaller scale, which is why we chose a range of 3 inlines and crosslines in our study. The same holds for the identification of karst structures, such as dolines, that are characterized by small-scale lateral and vertical variations, which is why small investigation windows were chosen. The investigation scale also




played an important role for the ant-tracking algorithm, because the presented workflow can also be used for the identification of large-scale faults and the extraction of corresponding fault patches. If the parameters are adjusted accordingly, for example, by changing the ant-tracking mode from aggressive to passive, and e.g. the initial ant boundary from a close distribution (2 voxels) to a coarse distribution (6 voxels). Therefore, it must be clear beforehand, what is to be highlighted with the attribute analysis, in order to select the appropriate parameters for the calculation.

Besides single-attribute analyses, the application of multiple seismic attributes in a combined plot is a key element of our study. They are generally usable in various clustering techniques, like self-organized maps (Roden et al., 2015; Zhao et al., 2015), geostatistics (Janson & Madriz, 2012; Ba et al., 2019), and neural networks (Brcković et al., 2017; Gogoi & Chatterjee, 2019; Abdel-Fattah et al., 2020) that have gained a lot of attention in recent years, because they enable parameter-based classifications, e.g., to obtain a 3D lithology or facies reservoir model. The quality of the results strongly depends on the input data such as seismic attributes and, in case of availability, the desired output data (e.g. lithology logs from wells). With regard to the input data, it is important to note that these must show a correlation between the physical parameters derived from the seismic attributes and the desired classes. A correlation coefficient close to 1 indicates a perfect or almost perfect match between the datasets. However, this is usually not the case for geological correlations, because rocks are generally heterogeneous, e.g., with regard to their petrophysical parameters or their composition. A correlation coefficient close to zero indicates that there is no relationship between the different datasets, which would make it impossible to achieve a good mathematical model that can be used for lithology prediction. Thus, only attributes with acceptable correlations should be implemented in the classification, because incorporation of attributes with low or no correlations would degrade the quality of the classification (Zhao et al., 2015). Therefore, we decided to choose only a small number of attributes that enable an acceptable classification. However, it should be kept in mind that the classification carried out in this study is based solely on different types of carbonates, which show minor physical differences. High correlation coefficients are therefore not to be expected. Besides the correlation analysis, we were also able to use a supervised neural network for the lithology classification, since six wells with corresponding lithology logs were available, which were used as the desired output data. Due to the fact that no large variations of the physical parameters between the different carbonate types were to be expected, a manual classification or an automatic classification with an unsupervised neural network that only looks for data trends would have hardly yielded promising classification results with a geological meaning. When using logs as boundary parameters, however, it should be noted that these should cover the entire range of the classes and therefore the heterogeneity of the reservoir, in order to enable a reliable assignment. As a consequence, the more heterogeneous the reservoir, the more wells should be implemented into the classification. With regard to our study, the implementation of only one or two wells might not be sufficient enough to obtain a representative lithology classification of the heterogeneously distributed carbonate types, e.g., Th5 contains mostly dolomitic limestone, Th2a contains mostly dolomite, and Th6 contains mostly limestone. Therefore, implementation of all six lithology logs delivered a more comprehensive overview of the different carbonate lithology types within the reservoir. Nevertheless, it is important to keep in mind that only a small part of the reservoir is covered by the six wells, which will always lead to a certain degree of uncertainty.

With respect to the identified classes, we chose to differentiate between the four carbonate types limestone, dolomitic limestone, dolomite, and calcareous dolomite, although the latter appears only sporadically. We chose this classification because





other studies have shown that, for example, dolomitization can have a major influence on reservoir quality, especially on poros-
ity and permeability (Reinhold, 1998; Koch et al., 2010; Böhm, 2012; Mraz, 2019; Bohnsack et al., 2020). There are other

classification options, but a classification, e.g., according to the two hyper-facies types (massive- and bedded facies), would
not have made geological sense, as many studies have shown that the presence of massive facies alone is no guarantee for good
reservoir conditions. For example, massive facies can also have poor poro/perm conditions due to, e.g., cementation, com-
paction and overdolomitization (Schmoker & Halley, 1982; Lucia, 2007; Wolfgramm et al., 2011; Homuth, 2014), whereas
bedded facies can also be suitable for exploitation, e.g., due to karstification along bedding planes and thus increased poro/perm

values (Lucia, 2007; Mraz, 2019; Wadas & von Hartmann, 2022). A further facies differentiation incorporating the lithology
types and also the fabric types (e.g. laminated, banded, ordered/unordered clasts, clast size) would be necessary, but since the
different facies classes did not show clear correlations with the seismic attributes because the rock-physics based parameters
did not vary strongly enough due to too many subdivisions, no reliable classification would be possible. Therefore, we limited
the classification to the lithology types.

It should also be noted that we used only volume attributes in this study but no horizon attributes because surface or horizon
attributes require that the horizons or surfaces were accurately picked. Regarding the Jurassic carbonates, this would only be
possible for the top of the reservoir. Internally, however, no horizons or surfaces could be picked due to the complex structural
conditions (e.g., the presence of faults, fractures, and dolines) and the complex lithology- and facies distribution (e.g., bedded
and massive facies). Seismic pre-stack attributes based on amplitude-versus-offset (AVO) analyses, for example, were also

not investigated, since an AVO analysis had not been carried out for the GRAME dataset at the time of this study and only
the finished stacked datasets were available. AVO attributes enable the offset-dependent investigation of physically-relevant
parameters of the reservoir and have provided promising results in other reservoir studies (Veeken, 2007; Barclay et al., 2008;
Bredesen et al., 2020). For this reason, we recommend that AVO analyses should be carried out together with the classical
post-stack attribute analysis in the exploration phase of future geothermal projects.

**5.2  Scalability of fracture orientations**

The comparison of fracture analyses from seismic and CMI data allows to draw conclusions regarding the scalability of frac-
tures in the GRAME area. As shown, the fracture orientations fit only under certain circumstances, thus, a general scalability
across the two scales, namely seismic- and well scales, is not given.

Possible reasons that fractures can not be scaled between seismic and drilling scales may be of geological, but also of

methodological origin. Regarding the geological reasons, von Hartmann et al. (2012), Lüschen et al. (2014) and Ziesch (2019)
have proven the existence of fault-related deformation for the Munich region, which according to our study seems to influences
scalability. Areas that are more affected by tectonic stress show poorer scalability than areas with less deformation, e.g. the
fault damage zones in the intermediate block and between the Munich Fault and the Nymphenburg Fault lead to intenser
fracturing, not only on seismic scale but also on sub-seismic scale (Lohr, 2008; Ziesch, 2016; Ashton et al., 2018). Therefore,

not all fractures in these areas are detectable by the seismic survey, especially if the fractures on the sub-seismic scale have
a different orientation than those on the seismic scale, resulting in non matching fracture clusters. Another control factor is



karst-related deformation. Dissolution of rocks can lead to cavities and over time new, especially small-scale, fractures can form in the surrounding areas due to collapse and local stress redistribution (e.g. Parise & Lollino, 2011; Salmiet al., 2017; Al-Halbouni et al., 2018; Shiau & Hassan, 2021). These fracture orientations do not have to follow the local stress field, but can

spread, e.g., radially away from the cavity (Schneider-Löbens et al., 2015; Rawal et al., 2016; Al-Halbouni et al., 2018). Thus, differences in the identified fracture orientations and fracture scalability can arise because these locally restricted fractures may not be detected on seismic scale, but they can be captured on CMI scale. However, the results of this study indicate that dolomitization is another factor that influences scalability, e.g., scalability decreases with increasing degree of dolomitization. During dolomitization, recrystallisation leads to a reduction in rock volume which can cause further fracturing and for that

reason, dolomite tends to have a higher fracture intensity compared to limestone (Korneva et al., 2018; Liu et al., 2020). To further validate these assumptions, additional analyses at other study sites are necessary. In summary, fracture scalability is possible, but depends much on the complexity of the geological conditions on sub-seismic scale.

Regarding the methodological reasons for different fracture orientations in seismic and CMI data, the different resolution limits of the two methods have already been mentioned. However, another important point is that a CMI analysis only covers

a very small volume, whereas 3D seismic attribute analysis allows a more comprehensive spatial analysis (Fang et al., 2017; Albesher et al., 2020; Boersma et al., 2020; Loza Espejel et al., 2020), and this may result in the detection of differing fracture orientations. In nature, fractures can change their orientation along the fracture path. Therefore, dip directions and dip angles can vary depending on where along the fracture path the orientation is measured. As a result, the values of the fracture orientations from the seismic data always represent an average value over the entire fracture path and not only a small part of the

fracture path, as is the case with the CMI data, although the CMI delivers a more accurate result for a specific investigation point. Therefore, the CMI may be less accurate in detecting large-scale structures than the 3D seismic, so the abundance distribution or intensity (n-value) of the preferred fracture orientations may be different based on the chosen method since different fracture sets may be captured. Furthermore, it also has to be considered that the n-value in the CMI and the seismic data can vary considerably, because, e.g., several parallel to sub-parallel fractures visible in the CMI data are imaged individually, whereas

the seismic data may only image them as a single fracture (with an averaged orientation) due to the seismic resolution. This means that a comparison of the n-values between seismic and CMI is not very meaningful. In addition, the different imaging of fracture orientations in the CMI data compared to the seismic data could be influenced by the well orientation, which introduces an orientational bias, e.g., fractures (sub-)parallel to the well can become underrepresented. With regard to the reflection seismic data, there are also methodical pitfalls that should be kept in mind. For example, not every discontinuity detected by

the seismic attributes has to be a real fracture, since it could be, e.g., an imaging artifact. Even if it is a real fracture, the derived geometric properties might be incorrect due to a wrong analysis window for the attribute calculation. For example, according to Marfurt & Alves (2015), coherence anomalies become increasingly more vertical the larger the analysis window is, resulting in a stair-step structure. Therefore, when analysing small features like fractures, small analysis windows must be chosen, as we have tried to do (see also details on the chosen attribute parameters in the appendix). Nevertheless, a small percentage

of uncertainty in the derived fracture geometries, which can not directly be quantified, must be expected. Another problem are possible lateral velocity variations of the seismic wave, inducing velocity pull-ups and push-downs that can cause false



structures in the seismic image. For example, high velocity anomalies result in a velocity pull-up and can generate subvertical 'fractures' (Marfurt & Alves, 2015). Furthermore, it must be considered that not all fractures, which are theoretically present in the seismic raw data, are captured by the attribute analysis. Processing steps, especially filters or smoothing algorithms that
are often used to enhance horizons, might remove real fracture information from the data unintentionally, instead of preserving/highlighting them, when processing parameters are chosen incorrectly. Another reason for the possible capture of too few fractures is scattering of the seismic wave at objects that are smaller then the wavelength. The resulting energy dispersion can then lead to fracture zones appearing as shadow zones in the seismic image and thus the fractures are not directly visible (e.g. Beilecke et al., 2016).

Overall, we conclude that both methods have their advantages and disadvantages. In our results we show that even with such a high data density in an comparatively small volume both methods produce results that are not generally comparable. However, we recommend to use both fracture orientation analysis methods. The seismic-based FOA is a useful addition to the CMI-based FOA, as it allows fracture orientations to be investigated at different scales and the combination of both analyses provides a more complete picture of the fracture inventory and local fracture orientation trends. This can also be helpful when
implementing fractures in other analysis methods such as hydraulic-mechanical modelling (Dussel et al., 2016; Bauer et al., 2019; Konrad et al., 2019). The seismic-based FOA can also be used to improve the planning of well paths. However, the seismic analysis cannot replace a later detailed CMI analysis, especially in areas with complex geological conditions as the Munich region.

## 5.3 Diagenetic and structural evolution of the reservoir below Munich

With the increasing economic use of the subsurface in the GMB, especially in the context of geothermal exploitation and the associated scientific exploration, more has become known about the geological development of the region. A better understanding of the geological evolution of the GMB, especially of the geothermal reservoir, is of great importance in order to identify controlling factors and to better estimate the exploitation potential. As already mentioned, the most important reservoir control factors in our study area are (I) the lithology and thus, the diagenetic evolution of the carbonates, (II) the distribution of massive
and bedded facies, (III) the tectonic evolution and the influence of faults and fault-related fractures, e.g. on fluid pathways, and (IV) the karstification processes that can also improve permeability through the formation of cavities and additional fractures due to stress redistribution. So far, these control factors have been determined primarily by well log analyses. 3D seismic data is not always available, and even when such data has been measured, it has been used primarily for structural geological interpretations in the past. However, 3D seismic data, particularly seismic attributes, also allow spatial interpretation of lithology,
diagenesis, and karstification as shown in this study. Therefore, in the following, existing concepts for reservoir development from spatially-constrained drilling studies in the GMB, e.g. cutting analyses and FMI analyses (Reinhold, 1998; Koch et al., 2010; Koch, 2011; Wolfgramm et al., 2011; Böhm, 2012; Beichel et al., 2014; Mraz, 2019; Bohnsack et al., 2020), are discussed, and an extended conceptual reservoir model for the Munich area is presented that also incorporates spatial variations based on the 3D seismic analyses carried out in this study.



A recent reservoir concept of the Upper Jurassic carbonates in the GMB, presented by Mraz (2019) and based on older works e.g. by Reinhold (1998), separates the diagenetic evolution of the Upper Jurassic carbonates into five phases: sedimentation, early diagenesis and first dolomitization, burial diagenesis and second dolomitization, late burial diagenesis, and the present-day reservoir. Taking into account the spatial analyses based on the seismic results of this study, it is concluded that, in addition to the different dolomitization phases, there were at least two karstification phases, and also various events that significantly

influenced the fluid pathways, whereby these processes also partially influenced or triggered each other. We have therefore expanded the reservoir development concept accordingly. Our extended reservoir development concept comprises the following steps: (1) sedimentation and reef development, (2) early diagenesis and first dolomitization, (3) burial diagenesis and second dolomitization, (4) subaerial exposure and widespread karstification and erosion, (5) Alpine orogeny and fault development, (6) late burial diagenesis and third dolomitization, (7) intense karstification along faults, and (8) continued subsidence (Fig. 15).

(1) Sedimentation and reef development: During the Upper Jurassic high amounts of carbonates were deposited in the study area that was covered by the Tethys Sea at that time (Schmid et al., 2005; Pieńkowski, 2008), which was also accompanied by the development of the reef buildups found in the GRAME area. As shown by the high-resolution acoustic impedance model derived from the GRAME 3D seismic by Wadas & von Hartmann (2022), the massive facies of the reef slopes interfinger with the surrounding bedded facies forming rounded 'pine-tree'-shaped edges. This was interpreted as an indication of syn-

sedimentary reef development with varying growth rates. The reef geometry also seems to narrow towards the top, especially in the upper third of the reef. This narrowing is postulated to result from long-term sea-level changes during the Upper Jurassic. According to Haq (2017), the sea level in the Oxfordian (Malm $\alpha$ to Malm $\beta$) and Kimmerdigian (Malm $\gamma$ to Malm $\epsilon$) rose from approx. +50 m to approx. +150 m (compared to today's sea level) and it dropped again to about +100 m until the end of the Tithonian (Malm $\zeta$). Since our study area was located in a shallow marine environment at that time, we assume that these

sea-level fluctuations had a significant influence on the carbonate production rates and thus also on the reef structure in the Munich area. A sea-level rise often enhances carbonate production, while a sea level fall can hamper carbonate production and therefore reef growth, especially in combination with enhanced erosion at the shallow coastline (Kendall & Schlager, 1981; Pomar & Ward, 1995). This is confirmed by the observation that the upper part of the reef buildups, mainly consisting of Malm $\zeta$, shows a decrease in lateral reef growth. Furthermore, the observed varying 'pine-tree'-shaped reef slopes (Wadas & von

Hartmann, 2022) might correlate with short-term sea-level changes. The fact that sea-level fluctuations have had an impact on our study area is important because they can also influence diagenetic processes, especially in shallow marine environments, e.g. at low sea-levels, freshwater diagenesis dominates, whereas at high sea-level, seawater diagenesis dominates, and if the area is completely exposed, erosion and karstification may also occur (Kendall & Schlager, 1981; Tucker & Wright, 1990; Bachmann & Müller, 1992). With regard to the further reservoir development, it is therefore important to know that, in general,

the sea-level fell further at the beginning of the Early Cretaceous down to +50 m and then rose again during the Cretaceous to up to +250 m, only to fall again in the Tertiary and Quaternary until it reached the present sea level (Haq, 2017).

(2) Early diagenesis and first dolomitization & (3) burial diagenesis and second dolomitization: The early burial diagenesis directly after deposition of the carbonates caused the first dolomitization phase and therefore the formation of matrix dolomite due to the contact with Mg-rich basinal fluids at 50 to 70°C (Mraz, 2019). This was directly followed by the second



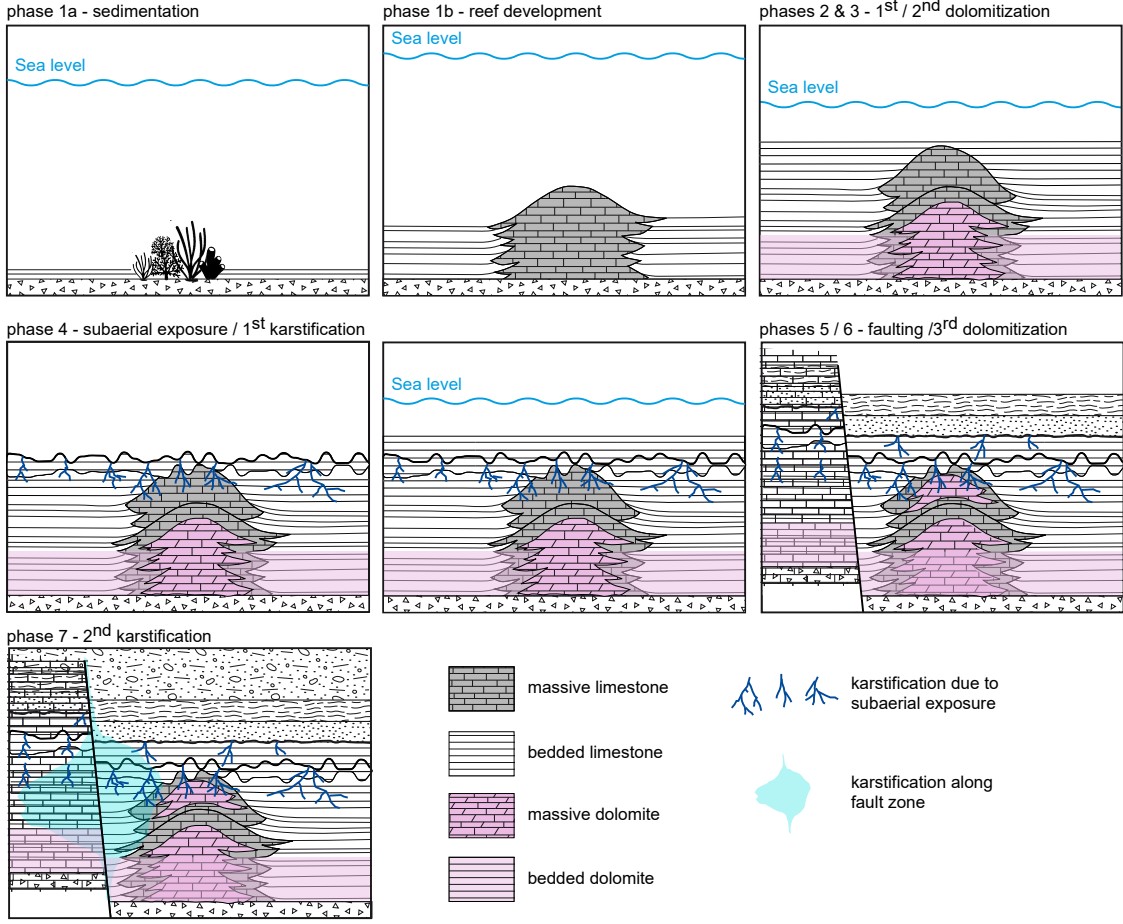

**Figure 15.** Schematic illustration of the reservoir development concept with its different phases: (1) sedimentation and reef development, (2) early diagenesis and first dolomitization, (3) burial diagenesis and second dolomitization, (4) subaerial exposure and widespread karstification and erosion, (5) alpine orogenesis and fault development, (6) late burial diagenesis and third dolomitization, (7) intense karstification along faults, (8) and further subsidence. Phases 8 is not shown.

dolomitization phase during burial diagenesis. According to several studies which investigated deep wells in the GMB and outcrops in the Franconian and Swabian Alb, the second dolomitization phase occurred at temperatures between 70 to 100°C and involved an enormous fluid flow, e.g. through highly permeable limestone (Lucia, 2007), with high-amounts of Mg-rich fluids causing intense dolomitization (Reinhold, 1998; Machel, 2004; Mraz, 2019). The fluids might have originated from sea water stored within the carbonate formations, which migrated through the permeable formations of the massive and bedded

facies during burial diagenesis, e.g., due to compaction-driven fluid flow (Reinhold, 1998; Machel, 2004). Our 3D lithology model (Fig. 12) shows clear spatial trends in the degree of dolomitization that can be assigned to the first two dolomitization phases and the later third dolomitization phase. Overall, the Upper Jurassic carbonates in the lower part of the reservoir are



more completely dolomitized than in the upper part, especially on the hanging-wall block to the south, creating a lithological
subdivision (Fig. 12). Furthermore, the footwall block shows a west-east to northwest-southeast trend, with higher degrees of
dolomitization in the west, and on the hanging wall a rough trend with more intense dolomitization in the central part and the
east can be seen. The intense dolomitization in the lower half probably results from the compaction-driven fluid flow of stored
Mg-rich sea water during burial diagenesis. The reason for the slightly stronger complete dolomitization in the south of the
study area could be that the former coastline of the Tethys was located to the north, and therefore the compaction pressure
towards the south, induced by sediment and water load, will have been slightly higher, which may have enhanced mobilization
of Mg-rich fluids. On the other hand, the west-east dolomitization trend on the footwall block and the east-west trend on the
hanging-wall block, seems to be more facies controlled. According to several studies, e.g., by Reinhold (1998); Mraz (2019),
massive facies (reefs) is more prone to dolomitization than bedded facies. As a result, completely dolomitized areas partly
correlate with reef buildups. Reefs are often characterized by grain-supported carbonates with biogenic components which are
mostly highly permeable (Flügel, 2010; Böhm, 2012; Homuth et al., 2015), at least during and shortly after deposition and
burial, enabling good conditions for enhanced fluid flow. This resulted in strongly dolomitized areas at the reef base and within
the reef cores, especially in the lower half of the reservoir.

(4) Subaerial exposure and widespread karstification and erosion: The falling sea-level during the Tithonian and especially
at the Jurassic/Cretaceous boundary led to subaerial exposure of the study site due to a regression of the Tethys. This enabled
widespread dissolution of the near-surface carbonate formations due to the contact with meteoric water (Bachmann et al.,
1987; Bachmann & Müller, 1992; Reinhold, 1998; Koch, 2011), which created a strong karst topography with many small-
scale dolines or sags, visible as blue-coloured dots in the shape index attribute (Fig. 11), covering the entire carbonate surface.
Dissolution and the associated karstification lead to the formation of secondary porosity, an enlargement of the already existing
pore space, and the formation of cavities. Furthermore, later cavity collapse can result in the formation of new fractures due
to local stress redistribution in the surrounding material, which is visible in the seismic attribute analysis by a strong scatter-
ing of the seismic waves at the doline edges, e.g. shown by low frequencies (Fig. 6) and high variance values (Fig. 9). Thus,
karstification can improve both the porosity and the permeability conditions of a reservoir. However, should the fluid solution
become supersaturated at some point, or in case of pressure and temperature changes, precipitation and thus cementation can
occur, which in turn can have a negative effect on porosity and permeability. The same goes for dedolomitization which is
often triggered due to contact with Ca-rich meteoric water, which enables the transformation of dolomite back into limestone
(Raines & Dewers, 1997; Reinhold, 1998; Koch, 2011), which in turn is usually accompanied by a deterioration in poro/perm
conditions. Since only the upper part of the reservoir was affected by Ca-rich meteoric fluids over a long time, no dedolomi-
tization is expected to have happened in the deeper parts of the reservoir (Mraz, 2019). However, since it is not possible to
distinguish between partially dolomitized limestone and partially dedolomitized dolomite in the reflection seismics, because
it is expected that the physical properties of both types would be the same, possible dedolomitization is therefore not further
discussed. Besides the karstification, another indication for the subaerial exposure at the Jurassic/Cretaceous boundary, is the
northeast-southwest oriented channel cut into the carbonate deposits, which can be seen in the acoustic impedance (Fig. 4) and



the spectral decomposition (Fig. 7). All these processes ended with the rise of the sea level during the Cretaceous, which again led to a flooding of the study area.

(5) Alpine Orogeny and fault development: The beginning of the Alpine Orogeny at the Cretaceous/Tertiary boundary
led to extensive compressional deformation, and therefore to the uplift of southern Germany (Lemcke, 1988). The tectonic deformation also led to faulting and the formation of fault-controlled fracture networks, which serve as fluid pathways for hydrothermal water, even in recent times (Mraz et al., 2018; Moeck et al., 2020). We analysed these fracture networks and their orientations in detail in the reflection seismic data and compared the seismic results with CMI results from the six wells at the 'Schäftlarnstraße' geothermal site. The results reveal W–E striking fractures parallel to the Alpine front, N–S striking fractures
perpendicular to the Alpine front and therefore parallel to $SH_{max}$, and NW–SE and NE–SW striking conjugate shear fractures. However, the fracture orientations, especially in the seismic data, show a wide range of orientations and it can be assumed that non-tectonic fractures, such as those caused by karstification and the collapse of cavities, also play a role in the fluid pathways and thus the permeability of the reservoir.

(6) Late burial diagenesis and third dolomitization & (7) intense karstification along faults: The fault and fracture-controlled
fluid flow also influenced the further development of the reservoir enabling a third dolomitization phase during late burial diagenesis, and a second karstification phase. Diagenesis continued especially in the Eocene and Miocene, where fault-controlled dolomitization occurred due to sea water migrating downward along the faults into convection cells (density driven circulation of fluids) where it was heated up and the Mg-rich hydrothermal water then migrated back upward along the fracture networks, enabling dolomitization in the younger carbonate formations (Benjakul et al., 2020). The upper part of the reservoir contains
mostly only partly dolomitized limestones, so we assume that either the fluid flow rates or the magnesium contents of the fluids were significantly lower than during the second dolomitization phase or even both together, which is why the dolomitization in the upper part of the reservoir area is less pronounced than in the lower part. Fluid migration of unsaturated water along the fractured fault zones was also the trigger for the second karstification phase. However, the dissolution processes and thus the karstification were concentrated along the fault zones and their immediate surroundings. As a result of the higher fluid flow
rates and the resulting intensified dissolution, significantly larger and deeper, and this time fault-related, dolines were formed, e.g., along the Munich Fault and the Ottobrunn Fault. Both the third dolomitization phase and the second karstification phase thus altered the poro/perm ratios of the reservoir again, creating more secondary pore space, especially in the upper part of the reservoir area, namely Malm $\zeta$.

(8) Further subsidence: The underthrusting of the European plate below the Adriatic-African plate in the Late Eocene caused
the Alpine nappes to extend northwards, which led to subsidence-induced development of the GMB (Frisch, 1979; Bachmann et al., 1987; Bachmann & Müller, 1992) and the formation of further faults, e.g. flexure-induced normal faults (Lemcke, 1988; Bachmann & Müller, 1992; Shipilin et al., 2020).

Overall, the Upper Jurassic carbonate reservoir experienced a complex evolution, consisting of at least three dolomitization phases, two karstification phases, and a phase dominated by tectonic deformation. We show that dolomitization in the GRAME
area is mainly facies-controlled and that karstification is both, facies- and fault-controlled. Karstification and cavity collapse



generally lead to improved porosity/permeability conditions for hydrothermal exploitation, while dolomitization can lead to both, an increase in secondary porosity, but also a possible porosity decrease due to overdolomitization.

## 5.4 Exploitation targets

Based on the comprehensive 3D seismic characterization of the Upper Jurassic carbonate reservoir, two regions are identified
that are likely to have suitable porosity and permeability conditions for future geothermal projects in the greater Munich area. The regions were selected according to the following criteria that are indicative of the important parameters permeability and porosity: lithology-type, degree of dolomitization, facies-type, karstification intensity, and fracture intensity.

The first recommended region is located in the southeast, at the Ottobrunn Fault. High porosities are expected here, according to the seismic inversion results by Wadas & von Hartmann (2022), due to partly dolomitized limestones and karstification along
the fault zones. Furthermore, the area also contains some reefs with good permeability conditions because of high fracture intensities caused by mainly fault-related deformation, but also mass redistribution at the reef slopes, and due to cavity collapse and doline formation. The second region is located north of the Munich Fault in the vicinity of the Nymphenburg Fault. High porosities are caused by partly dolomitized limestone and widespread karstification. In contrast to the first region, this area is located in mostly bedded facies and the karstification does not seem to be fault-controlled because no significantly large dolines
are observed along the Nymphenburg Fault. Instead mainly widespread karstification with large clusters of very small-scale dolines are observed. The widespread karstificatiion and the influence of the fault zone also resulted in the generation of a presumably highly permeable reservoir.

## 6 Conclusions

With our case study from the South German Molasse Basin, we show that a comprehensive seismic attribute analysis can
significantly improve the understanding of complex carbonate reservoirs. We addressed the physical and structural characterization of the reservoir, the question of the scalability of fracture orientations on seismic and well scale, the creation of a 3D lithology model, the temporal development of the reservoir, and the identification of further suitable exploitation targets in the GRAME area.

For the carbonates in the Munich area, we summarise that amplitude-related attributes, especially the acoustic impedance,
are suitable to identify reefs due to the strong amplitude contrasts to the surrounding material. Sweetness has proven to be a useful tool to analyse the internal reef architecture. The frequency- and phase-related attributes, on the other hand, are well suited for the detection of karst, since the high frequencies are attenuated especially in fault- and karst-related fracture zones. In addition, the edges of reefs, dolines, and fractures are characterized by strong phase changes. Fractures can also be identified using discontinuity-related attributes, such as variance and ant tracking. Furthermore, morphological characteristics like bowl-
shaped dolines are captured using the shape index (see also appendix table B1, with a tabular overview of seismic attribute analysis results).



Regarding fracture scalability across seismic and well scale, we state that a general scalability across the two scales is not possible, although the fracture orientations are partially in good agreement. Limiting factors are methodological differences between seismic-based FOA and CMI-based FOA, as well as complex structural and lithological conditions, such as fault- and

karst-related deformation on subseismic scale and the degree of dolomitization. Nonetheless, seismic-based FOA is a useful addition to the CMI-based FOA, as it allows fracture orientation to be investigated at different scales and the combination of both analyses provides a more complete picture of the fracture inventory and local fracture orientation trends, which can improve hydraulic-mechanical modelling results.

We also created a 3D lithology model using parameter relationships between the seismic attributes and the carbonate litholo-

gies, based on a neural network. Together with the physical- and structural reservoir characterization, this enabled a closer delineation of control factors, indicating good target areas for geothermal exploitation. In terms of lithology, the dolomitized limestone should be targeted due to increased secondary porosity caused by the partial transformation of limestone into dolomite and because they are more prone to karstification than fully dolomitized carbonates. Dolomite rocks should even be avoided because they tend to be of low-porosity in the GRAME area, possibly due to overdolomitization, which can negatively

affect porosity and permeability. Therefore, only those reservoir zones should be exploited which have not been affected by too strong diagenetic alteration. This concerns mainly the upper half of the Upper Jurassic carbonates with the Tithonian deposits. With regard to karstification, we conclude that it can lead to good porosity/permeability conditions for geothermal exploitation. As preferred karstification areas, we can identify the upper part of the reservoir due to a past suaberial exposure at the Jurassic/Cretaceous boundary, and a later intensive karstification along faults related to the fault evolution during the Alpine

orogeny and the development of the foreland basin. Taking all the results into account, we are able to identify two more regions in the Munich area that are likely to have preferable conditions for future geothermal exploitation projects; the region in the southeast at the Ottobrunn Fault and the region north of the Munich Fault in the vicinity of the Nymphenburg Fault.

Based on the promising results of this study, we recommend that in future geothermal exploration projects in the GMB, but also in other areas, much more comprehensive seismic analyses should be carried out, in contrast to previous studies,

where the focus was mainly on the interpretation of faults and marker horizons and, if necessary, the localization of reefs. This comprehensive seismic analyses includes methods, such as attribute analyses, e.g., for the description of fracture zones, karstified areas, and the structural composition of reefs and dolines, inversion methods to calculate a high-resolution impedance model and to estimate a 3D porosity model, lithology and facies classifications, and seismic fracture orientation analyses. The knowledge gained can help to develop a better understanding of the reservoir, its evolution and the distribution of relevant

exploitation parameters. Furthermore, the obtained data can also serve as input parameters for further studies, such as dynamic modelling, which strongly depend on the defined constraints. For example, when modelling stress changes in the uppermost crust, the associated possible reactivation of faults, and the propagation of fractures, during the theoretical operation of a geothermal doublet with a production- and an injection well, it is important to know how the fracture inventory looks like. Information on fracture densities and also fracture orientations are also important in the context of thermal-hydraulic modelling,

as fractures represent fluid pathways of hydrothermal waters. Here, a precise knowledge of the spatial distribution of certain lithology and facies types, but also karstification zones is also important for realistic modelling approaches.



*Data availability.* The data that support the findings of this study are available from the Stadtwerke München (SWM) and the Leibniz Institute of Applied Geophysics (LIAG), but restrictions apply to the availability of these data, which were used under license for the current study, and so are not publicly available. Data are, however, available from the authors upon reasonable request and with permission of SWM
and LIAG.

**Appendix A: Methods**

**Amplitude-related attributes**

Three often used amplitude-related attributes are the RMS amplitude, the reflection intensity and the envelope. The RMS amplitude is described as the square root of the sum of squared amplitudes divided by the number of samples within a specified
time window (Chopra & Marfurt, 2007). The default window length in Petrel is 9 samples (Schlumberger, 2020), but since we expected very small-scale variations of the energy content of the seismic data, e.g. at dolines and reef buildups, a window length of 5 samples was chosen after testing different windows. Another amplitude-related attribute that can help to distinguish between different lithologies, and which was tested in this study, is the reflection intensity (Chopra & Marfurt, 2007; Sarhan, 2017), which is the average amplitude over a specified time window (5 samples) multiplied with the sample interval. And
the envelope, also known as the instantaneous amplitude, is the magnitude of the complex trace described by $Envelope = \sqrt{f^2 + g^2}$ where $f$ and $g$ are the real and imaginary components of the seismic trace. It is independent from phase and can therefore highlight lithological changes, sequence boundaries, and thin-bed tuning effects (Chopra & Marfurt, 2007). The default window length in Petrel is 33 samples (Schlumberger, 2020), but we chose 15 samples in order to depict small-scale variations.

As shown in the appendix figure A1 these three attributes can highlight areas with a much higher energy content compared to their surroundings, as it can be seen, e.g., in the western part of the footwall block (Fig. A1a, d & g) of the Munich Fault and the eastern part of the intermediate block (Fig. A1b, c, e, f, h & i). Such a strong difference in the amplitude-, intensity- or envelope values can be an indicator for lithological, diagenetic and/or depositional changes that effect the rock properties. Therefore, the larger but locally restricted high amplitude anomalies on the footwall and the intermediate block are presumed to
result from reef buildups. Such high amplitude values for reefs are also observed by other studies, e.g. Sarhan (2017). But it has to be noted that since RMS amplitude, reflection intensity, and envelope mainly identify strong anomalies they are unsuitable to depict small variations in energy content, e.g. due to karst-related dolines, especially on a small area, as it is the case in our study area. In general, this can hamper a sophisticated identification of structures/features and a detailed interpretation, e.g. of the internal architecture of the carbonate features.

One attribute that allows the identification of small variations is the acoustic impedance (Fig. 4). Every reflection changes the amplitude of the returning wave due to a contrast in acoustic impedance, which is the product of the seismic velocity of the wave travelling through the subsurface and the density of the rock (Barclay et al., 2008). Therefore, the reflection amplitudes can be used to invert the data to get impedance values by using e.g. a stochastic seismic amplitude inversion as carried out for our study area by Wadas & von Hartmann (2022). The advantages are that (1) the calculated impedance model delivers a



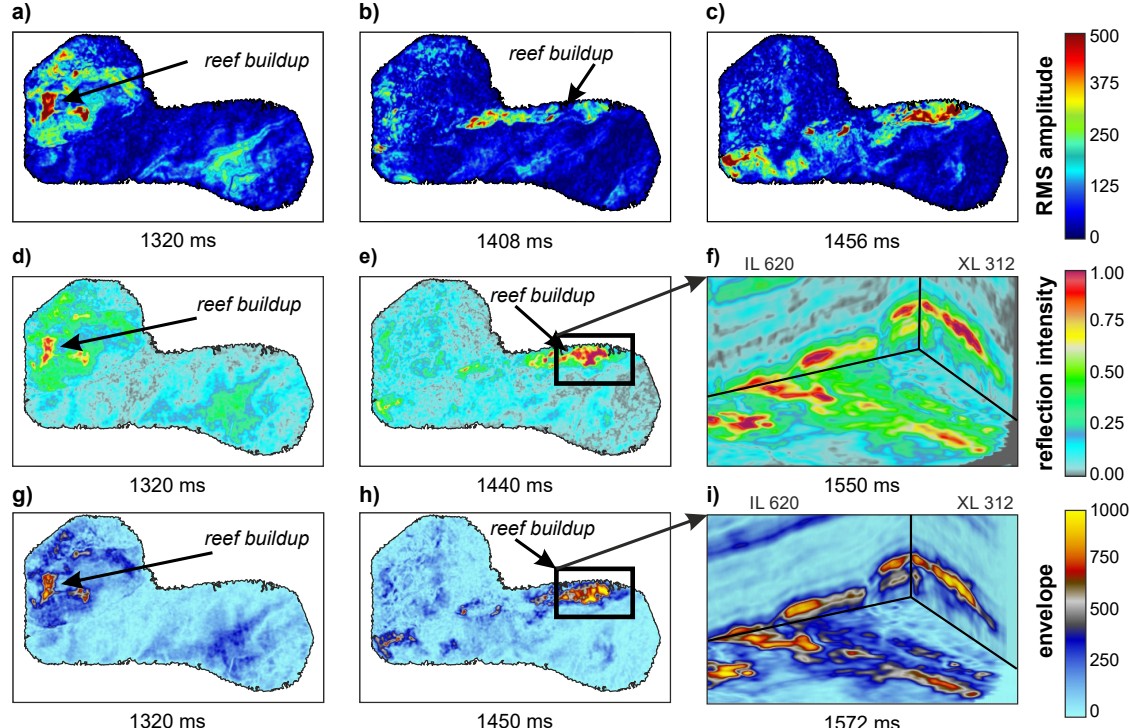

**Figure A1.** Time slices and zoom-ins of the RMS amplitude volume (a-c), the reflection intensity volume (d-f) and the envelope volume (g-i). The time slices show locally restricted high amplitude anomalies interpreted as a possible reef buildups.

.

subsurface image with higher resolution due to a reduced tuning effect (Hill, 2005), which allows a more detailed structural and lithological interpretation compared to the classical amplitude attributes, (2) the impedance data can be calibrated with well log data, (3) the data has an increased bandwidth due to implementation of frequencies beyond the seismic bandwidth, e.g. from well logs, (4) it enables the integration of horizons and geological structures, and (5) the derivation of reservoir parameters based on a strong relationship between acoustic impedance and petrophysical properties (Pendrel & Van Riel, 1997; Hill, 2005; 910    Sarhan, 2017; Gogoi & Chatterjee, 2019).

**Phase- and frequency-related attributes**

The instantaneous phase (Fig. 5) of the seismic data is calculated by $Phase = \arctan(\frac{g}{f})$, where $f$ and $g$ are the real and imaginary components of the seismic trace. It measures the phase shift of a specific reflection event, e.g. resulting from a polarity reversal of the reflection coefficient or due to a curved interface (Chopra & Marfurt, 2007). It is calculated sample by sample 915    without regard of the waveform over a specified window. The default window length in Petrel is 33 samples (Schlumberger, 2020), but we chose 15 samples in order to better depict small-scale variations.



The instantaneous frequency is the rate of change of the instantaneous phase from one time sample to the next, $Inst. frequency = \frac{Delta\ phase}{Delta\ time}$, over a specified window. The dominant frequency (Fig. 6) is the square of the instantaneous frequency summed with the square of the instantaneous bandwidth and then the square root of the sum is calculated (Schlumberger, 2020). So

the dominant frequency indicates where the energy of the seismic signal is concentrated in the frequency domain. The chosen window length for both attributes was 15 samples in order to depict small-scale variations. Both frequency attributes are good indicators of bed thickness and changes caused by e.g., due to fault and fracture zones or differing lithology (Van Tuyl et al., 2018), by distinguishing low and high frequency areas (Sarhan, 2017).

**Discontinuity-related attributes**

For the variance (Fig. 9) a trace-by-trace analysis is performed in order to quantify the dissimilarity of the seismic waveform of neighboring traces within a specified time window (Bahorich & Farmer, 1995; Marfurt et al., 1998; Wang et al., 2016). The default inline and crossline ranges in Petrel are 3 and we chose a range of 2 in order to enhance small-scale dissimilarities. For vertical smoothing only a mild smoother with 12 samples was applied to prevent smearing of the detected edges and, since we wanted to enhance not only faults but also stratigraphic and karstic features, no dip correction/guidance was applied. Chaos

maps the "chaoticness" of the seismic signal from statistical analysis of dip/azimuth estimates. With directional sigma values the user can define the window radius for calculating the chaoticness of the seismic data and the larger the sigma, the smoother the result. Therefore, in order to depict small-scale discontinuities a value of 1.0 was chosen for Sigma X, Sigma Y and Sigma Z (Schlumberger, 2020). Both, variance and chaos, are commonly utilized to visualize geologic features that are characterized by lateral discontinuities, such as stratigraphic terminations or structural lineaments like faults, fractures, dolines, and reef

edges (Chopra & Marfurt, 2007). Since both attributes deliver similar results, only variance is shown (Fig. 9).

**Sweetness**

Sweetness (Fig. 8) is the combination of instantaneous frequency and envelope and is described by $Sweetness = \frac{Envelope}{\sqrt{Instantaneous\ frequency}}$ (Radovich & Oliveros, 1998). High sweetness values are correlated with both high envelope and low instantaneous frequencies, whereas low sweetness values result from low envelope and high instantaneous frequencies.

**Spectral decomposition**

Spectral decomposition (Fig. A2) separates the seismic signal into its frequency components. Three different frequencies are then co-rendered and displayed by RGB colour-blending, where frequency 1 is plotted against red, frequency 2 is plotted against green and frequency 3 is plotted against blue. Since this is an additive colour model, three equally strong signals of the frequency components will result in a white response (Chopra & Marfurt, 2007; Al-Maghlouth et al., 2017). In Petrel a

hybrid method, combining a Short Time Fourier Transform (STFT) and a Continuous Wavelet Transform (CWT), is applied that allows to control both the vertical- and the frequency resolution (Schlumberger, 2020). After iteratively testing different



frequency components and their combinations, 18 Hz, 28 Hz and 50 Hz were chosen as the best suited frequencies and the number of cycles was 3 (a small number of oscillations will deliver a better vertical resolution due to the short wavelet).

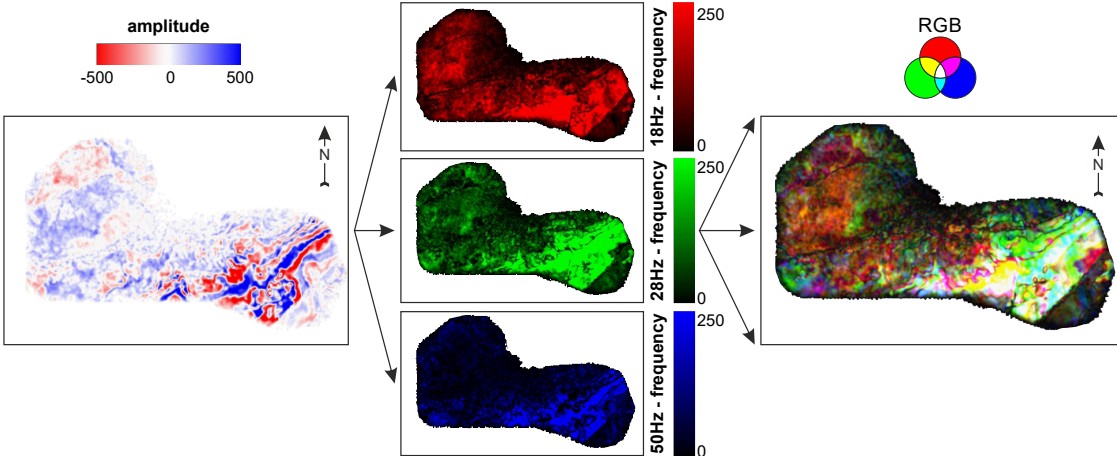

**Figure A2.** Spectral decomposition of the GRAME data with 18 Hz plotted against red, 28 Hz plotted against green and 50 Hz plotted against blue.

**Curvature**

Curvature (Fig. 10) measures how much a seismic reflector is bent. In case of a planar reflector, the corresponding vectors are parallel and as a consequence the reflector has zero curvature at this location. In contrast, anticlinal features result in diverging vectors and the curvature is defined as positive and synclinal features result in converging vectors and the curvature is defined as negative (Roberts, 2001; Al-Dossary & Marfurt, 2006; Wang et al., 2016). Different curvature types that can be calculated are, e.g., gaussian curvature, mean curvature, minimum/maximum curvature, and most positive/most negative
curvature. According to Chopra & Marfurt (2007), the latter are the most suitable to visually correlate with geologic features. The most positive curvature is described as the most positive value of all possible normal curvatures at a point, and the most negative curvature is defined as the most negative value of all possible normal curvatures at a given point. The search radius of the algorithm must be selected depending on the research question/the object to be investigated (Roberts, 2001), e.g. a large search radius is needed for detection of regional trends (long wavelength structures) and a small search radius is necessary for
small local features (short wavelength structures). Since in our study especially small-scale structures like buildups and dolines are to be investigated a small search radius was chosen with a vertical radius of 10 samples (the window size is two times the vertical radius plus 1) and an inline/crossline radius of 2 (Schlumberger, 2020).





**Shape index & Curvedness**

Formula to calculate the Shape Index $\rightarrow S = \frac{2}{\Pi} * \tan^{-1} \frac{K_{min} + K_{max}}{K_{min} - K_{max}}$ (Al-Dossary & Marfurt, 2006).

Formula to calculate the Curvedness $\rightarrow C = \sqrt{\frac{K_{max}^2 + K_{min}^2}{2}}$ (Al-Dossary & Marfurt, 2006).

**Ant-tracking**

In the following Schlumberger's patented ant tracking algorithm and the chosen parameters are described in detail: The initial ant boundary that defines the initial search radius of the agents by a number of voxels (a voxel is defined as the volume around one sample, which in our case relates to the sample rate of the variance cube) in which the agent tries to find a local maximum

was set to 2 voxels. So if the agent is unable to find a local maximum within this radius, the agent will be removed. This close distribution enabled the mapping of small fractures. With the ant track deviation parameter the maximum allowed deviation while tracking can be determined. For example, a value of 2, as chosen in our study, enables the agent to deviate by 2 voxels in every direction in order to find the next local maximum (values between 0 and 3 voxels possible, default is 2). If none is found this would be considered an illegal step. And with the ant step size the user can define the number of voxels the ant agent can

advance during each searching step. So large vaules allow the agent to search further away for a local maximum, although this will reduce the resolution of the resulting ant tracking volume. Since our aim is to get a high resolution volume the step size was set to 2 voxels (values between 2 and 10 voxels possible, default is 3). The user can also specify the number of allowed legal and illegal steps. The legal steps parameter describes the number of steps that must contain a valid edge value for the agent to continue. For example, when the agent encounters a valid edge (local maximum), this would represent the 1st legal

step. When the agent continues its search and finds another valid edge this would be the 2nd legal step. In case the user has set this parameter to 2 legal steps (values between 0 and 3 steps possible, default is 3 for passive mode and 2 for aggressive mode), as it was done in this study, this ant track would be considered as legitimate and the agent can further continue. If the parameter would have been set to 3 legal steps and no edge is found after the 2nd legal step the track will not be forwarded to the final ant track volume. The illegal steps parameter, which was set to 2, defines how far the ant track can continue without

finding a local maximum (values between 0 and 3 steps possible, default is 1 for passive mode and 2 for aggressive mode). An agent can accumulate illegal steps over time until they can represent a significant portion of the entire ant track leading to uncertainties. To prevent this, a stop criteria given in percent can be defined, which automatically terminates the ant track when the illegal steps exceed the given percentage. In our study we chose a stop criteria of 10% (values between 0% and 50% possible, default is 5% for passive mode and 10% for aggressive mode). For the 2nd ant track the same parameters were chosen

as for the 1st ant tracking, except for the initial ant boundary which was set to 4 voxels. Afterwards, automatic fault/fracture extraction (Schlumberger, 2020) was used to extract 3D fracture patches from the ant-track volume. The extraction sampling threshold, which sets the minimum signal level from which extraction points were created, was set to the top 30% of the data, meaning that only the highest ant track values, the sharpest edges, were used for the patch extraction process. Furthermore, the minimum patch size was set to 100 connected voxels.



**995  Appendix B:  Recommended seismic attributes**



|  | SEISMIC ATTRIBUTE | DESCRIPTION | APPLICATION | Reef | Karst | Fault/ Fracture |
|---|---|---|---|---|---|---|
| SINGLE-ATTRIBUTES | RMS amplitude | Square root of the sum of squared amplitudes divided by the number of samples within a specified window. | Used as an indicator for strong lithological and/or depositional changes, e.g. reefs are characterized by high values, but is unsuitable to depict small variations in energy content. | ✓✓ |  |  |
|  | Reflection intensity | Average amplitude over a specified window multiplied with the sample interval. | Used for identification of strong lithological contrasts, e.g. reef buildups, similar to RMS amplitude. | ✓✓ |  |  |
|  | Envelope | Magnitude of the complex (or analytic) trace, independent from phase, also known as instantaneous amplitude. | Highlights strong amplitude contrasts and sequence boundaries, e.g. reef buildups, by high values. | ✓✓ |  |  |
|  | Acoustic impedance | Product of seismic velocity and density, derived from seismic inversion. | Delivers a subsurface image with a seismic-like lateral and a sonic log-like vertical resolution, which allows a more detailed interpretation of e.g. reefs, karst and channels. Low values can also be an indicator of high porosity. | ✓✓✓ | ✓✓ |  |
|  | Instantaneous phase | Measures the phase shift of a reflection event, e.g. resulting from a polarity reversal of the reflection coefficient or due to a curved interface. | Used to find continuity of weak reflectors and identify stratigraphic sequences, and boundaries, like reef edges and doline margins. | ✓✓ | ✓✓ | ✓ |
|  | Instantaneous frequency | Rate of change of instantaneous phase from one time sample to the next (first derivative of the phase). | Strong damping of high frequencies due to scattering at small-scale edges can be an indicator of karstified and/or fractured areas. | ✓ | ✓✓ | ✓ |
|  | Dominant frequency | Sums the square of the inst. frequency with the square of inst. bandwidth and calculates the square root of the sum. | Indicates where the energy of the signal is concentrated in the frequency domain and highlights changes in geology, e.g. due to damping of high frequencies at fault and fracture zones or karstified areas. | ✓ | ✓✓ | ✓ |
|  | Chaos | Maps the "chaoticness" of the seismic signal from statistical analysis of dip/azimuth estimates. | Utilized to visualize lateral discontinuities, such as stratigraphic terminations or structural lineaments like faults, fractures, dolines, and reef edges which show high chaos values. | ✓ | ✓ | ✓✓✓ |
|  | Variance | Variance quantifies the dissimilarity of the seismic waveform of neighboring traces within a specified time window on a trace-by-trace basis. | High variance values indicate high dissimilarities due to e.g. the presence of strong fractured rocks caused by e.g. fault- and karst-related deformation. | ✓ | ✓✓ | ✓✓✓ |
|  | Edge evidence | Enhances the edges in a discontinuity volume. | Used to highlight faults and fractures. |  |  | ✓✓✓ |
|  | Ant tracking | Simulates the behaviour of ants which use pheromones to optimize their search for food by marking their paths. | Artificial ants are used to search for faults and fractures generating an attribute volume with very sharp edges. |  |  | ✓✓✓ |
| MULTI-ATTRIBUTES | Spectral decomposition | Separates the seismic signal into its frequency components. | Useful tool to enhance subtle structural features like thin beds, reefs, channels, and it can also be used for seismic geomorphology analysis. | ✓✓ | ✓✓ | ✓ |
|  | Sweetness | Combination of instantaneous frequency and envelope. | It is able to detect general energy changes and it can enhance not only lithology, but also e.g. discontinuities, stratigraphic changes, and depositional architecture, and is therefore useful to characterize the reef interior. | ✓✓✓ | ✓ |  |
|  | Curvature | Measures how bent a curve is at a specific point on the curve. Curvature is defined as the rate of change of direction of a curve. Anticlinal features are assigned a positive value und synclinal surfaces a negative value. | Can be used to identify fractured zones, faults and also dolines. | ✓ | ✓✓ | ✓✓✓ |
|  | Shape Index | Calculated based on curvature and describes the shape of local structure as dome, bowl, ridge, plane, valley and saddle. | Ideally suited to depict even small-scale dolines, but also faults and fractures. |  | ✓✓✓ | ✓✓ |
|  | Curvedness | Calculated based on curvature. | Measures the intensity of folding, e.g. a value of zero indicates a plane surface. |  | ✓✓ | ✓✓ |

**Table B1.** Recommended seismic attributes for the detection and parametrization of controlling factors in a deep geothermal carbonate reservoir, such as reefs, karst, faults and fractures.



*Author contributions.* The data that support the findings of this study was provided by the Stadtwerke München (SWM). Seismic data analysis was carried out by SW and image log analysis was carried out by JB. Interpretation of the seismc analysis results was jointly carried out by all auhtors. The interpretation and comparison of the fracture orientation results on seismic and well scale were done by SW, JB and MK. SW prepared and discussed the results with all co-authors. All authors have contributed to the writing of the manuscript, and SW and JB created the figures.


*Competing interests.* The authors declare that they have no competing interests.

*Acknowledgements.* We thank the German Federal Ministry of Economic Affairs and Energy for the financial support of the project REgine (project number 0324332B), which is part of the joint project GeoMaRe. Furthermore, we thank the Stadtwerke München (SWM) for the cooperation and for the seismic data, without which this study could not have been carried out.





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
