# Peer review of "Advanced seismic characterization of a geothermal carbonate reservoir – Insight into the structure and diagenesis of a reservoir in the German Molasse Basin"

_EGUsphere, 2023_

## Author Response (AR1)

| Point | Chapter | Reviewer | Comment | Agreement/ disagreement | Answer |
|---|---|---|---|---|---|
| 1 | General comments | R2 | The paper is well written and well-structured. The work is based on data of high quality. Data analysis is carried out with existing state-of-the-art methods. Figures are of excellent quality. The results are described and discussed regarding both methodological aspects and geological interpretation. | / | We are pleased that reviewer R2 enjoyed reading our manuscript and we want to thank the reviewer for the useful comments and suggestions that helped to improve the manuscript. |
| | Abstract | | None | | |
| 2 | 1 - Introduction | R2 | The motivation for the study is reasonably explained by the very complex situation  and specific challenges of carbonate reservoirs for geothermal exploration. | / | We thank the reviewer for a positive assessment of our introduction. |
| | 2 - Study site | | None | | |
| 3 | 3 - Methods | R1 | For the supervised neural network, what is the network structure used in the study? | Agreed | We thank the reviewer for this useful comment and added more information about the neural network (see lines 245 to 253 in NEW manuscript version with tracked changes). |
| 4 | | R1 | In the confusion matrix, people have to check every element in the matrix to analyze the prediction accuracy. It is better to provide an overall/single index for an easy analysis. | Partly agreed | The confusion matrix is a standard technique for summarizing the performance of classification algorithms. It's advantage is that it gives the classification accuracy and also lists what the classification model gets right and what it gets wrong. Giving, as requested, only an overall single index per class would therefore lead to a significant loss of information. Both reviewers, however, have asked for more information regarding the neural network. A reduction of the information regarding the validation of the neural network is therefore a contradiction to the desired increase in information. Nevertheless, we understand that not every reader is familiar with neural networks, and therefore we added further information in the figure caption and adjusted the figure a little bit to improve the understanding of the confusion matrix table. |
| 5 | | R1 | For the lithology prediction, what are the inputs for the classification? | Partly agreed | As input data for the lithology classification we used six seismic attribute "logs" (acoustic impedance, dominant frequency, reflection intensity, variance, envelope, and the 28Hz frequency band) derived from the seismic attribute volumes along the well paths. These six attributes show good correlation with the desired output data (the lithology logs). The same seismic attributes were then used for the generation of the 3D prediction model. This information is given in lines 225 to 234, and line 249 to 250 of the originally submitted manuscript. To clarify this point, we added "To get input parameters for the neural network..." in front of the aforementioned paragraph. |
| 6 | | R2 | The underlying data are described at the beginning of the Methods. The authors could think about a separate chapter to describe the Data. But the presented version is also clear enough with given references for the data. | Agreed | According to the reviewer's suggestion, we have moved the information about the seismic dataset from the beginning of the "Methods"-Chapter into a newly inserted "Database"-Chapter. In addition, we give information on seismic data processing and the borehole data. |

| 7 | | R2 | The advanced data analysis is nicely categorized into four seperate approaches:
+ single attribute analysis
+ multi-attribute analysis
+ neural network-based lithology classification
+ fracture orientation analyis
I suggest to extend a bit the explanation of the neural network-based clasification.
My assumption at this point is, that a few readers would like to know more technical details such as
+ type of neural network
+ architecture
+ specifications of network (e.g. learning rule, internal functions)
+ software used / implementation | Agreed | As suggested by the reviewer, we have added more information on the neural network (see lines 245 to 253 in NEW manuscript version with tracked changes). |
|---|---|---|---|---|---|
| 8 | 4 - Results | R1 | In the analysis of the seismic cube for internal structure of the GRAME area, different subareas have been chosen in different technologies (Figs. 4-11, such as a,b,c,d,e). Could you focus on the same area and apply the different seismic data analytic methods for a consistent evaluation? | Not agreed | We understand the request of the reviewer, but the combination of the large size of the investigated area and the heterogenous spatial distribution of the comparatively small structures (e.g. dolines with a few tens of meters in diameter) that in addition can have strongly contrasting characteristics did not allow for such an approach. For example, it is useful to show in Figure 6 (which deals with the dominant frequency), the reef in the west of the hanging wall, because a clear difference in the frequencies can be seen compared to the reef in the east of the intermediate block. But since there was no significant difference between these two reefs in e.g. the phase-attribute (Fig. 5), only the reef in the east of the intermediate block was shown in that figure.

Furthermore, as described in detail in the Methods-chapter, the different attributes show different seismic parameters and not each of these parameters can give information about the same geological structures or features or facies nor at the same scale.

We also want to point out that we already show similar sections in the zoom-ins of the different seismic attributes. For example, the zoom-in on the reef to the east of the intermediate block can be seen in Figures 4b, 5d, 6b, 8b, and 9e. The zoom-in on the dolines at the Munich Fault are shown in 4c, 5b, 6c, 9d, and 10d, and the zoom-in on the dolines at the Ottobrunn Fault are shown in 4e, 5c, 6d, 9c, 10e, and 11e.
In addition, we mention that reviewer R2 praised the figures as excellent. |
| 9 | | R2 | The important support for the interpretation of results in this paper is given by the presented empirical correlations between attributes and borehole data. The compehensive attribute analysis and their interpretation is a very nice contribution to improve the characterization of geothermal carbonate reservoirs. | / | We thank the reviewer for the compliment. |

| | | | | | |
|---|---|---|---|---|---|
| 10 | 5 - Discussion | R1 | Please rewrite the discussion section, as some new figures and equations are introduced. And also the discussion should be more focused on the main findings and limitation of the current research, as well as some suggestions for future researches. | Not agreed | We disagree with the reviewer's suggestion because it seems to us that the reviewer has mixed up the discussion and the appendix, since the discussion contains no equations and only one new figure, which is explained in detail in Chapter 5.3. In contrast, the appendix, in which some of the methods are described in more detail, actually contains several figures and equations. However, the appendix does not belong to the discussion.

Regarding the comment that we should focus on the limitations of our research: In the Discussion-Chapters 5.1 (Methodical approach) and 5.2 (Scalability of fracture orientations) we already outlined in detail the limitations of the applied methodical approaches and therefore our results. Reviewer R2 has also written that "the critical discussion of the seismic attributes as given in the discussion chapter" is commendable.
Regarding the comment that we should make suggestions for future research: We already give explicit suggestions for methodological improvements in exploration, e.g. in the last paragraph of Chapter 5.1, and we also make suggestions with regard to exploitation and name possible targets for future geothermal projects in Munich in Chapter 5.4 (Exploitation targets). |
| 11 | | R2 | Commendable for me in this manuscript is the critical discussion of seismic attributes as given in the discussion chapter. Maybe it would be worth to mention that seismic attributes are signal properties, and not inherent rock physical properties.
As an example, frequency based attributes could be influenced by different factors including inherent seismic attenuation, complex geological structures such as thin layers or gradient structures with potential shifts of signal frequencies, data processing or acquisition footprints.

At least from my perspective the formulation "reservoir control factors that may affect the physical properties of the seismic signal" (line 550) is suggesting that the signal properties of the seismic reflection waveform directly represent subsurface rock physical properties. As descibed above, the causality is more complicated in my opinion.
But this might be an overcritical comment. | Agreed | Regarding the definition of seismic attributes and the corresponding formulation in line 550: We followed the advice of the reviewer and clarified this point in the discussion (Chapter 5.1) and have now written that seismic attributes are properties of the seismic wave, e.g. amplitude, frequency, attenuation,.... We have also incorporated that acquisition footprints, processing artefacts, and noise can negatively influence the quality of the attribute analysis. For that reason, a quality check of the seismic data should be done in advance. |
| 12 | 6 - Conclusions | R1 | It is also for the conclusion which is too long. Please shorten it. | Agreed | As suggested by the reviewer, we have shortened the conclusion and it is now 12 lines shorter. |
| | Appendix | | None | | |
| 13 | Figures & Tables | R1 | In figure 2d, what is the gray color in Th3? | Agreed | We thank the reviewer for noticing this mistake and we corrected the legend. The grey colour marks areas where lithology information from the wells is not available due to heavy mud loss. Furthermore, we improved Fig. 2. |

| 14 | References | / | / | / | Although there was no reviewer comment regarding the references, we have checked the reference list and corrected it in order to meet the journal standards. |
|----|-----------|---|---|---|---|
| 15 | Spelling mistakes | / | / | / | We corrected spelling mistakes and small errors that were not requested by the reviewers. These small changes can be traced in the manuscript version with tracked changes. |

---

## Author Response (AR2)

**Cover letter for 2nd paper resubmission**

**Author response to comments by the topical editor**

We would like to thank the topical editor, Ulrike Werban, for her positive evaluation of our manuscript and we adjusted the abstract according to her suggestions.

The manuscript with highlighted changes (tracked changes) can be found on the following pages.

With best regards,

Sonja Halina Wadas - on behalf of all co-authors

[revised manuscript text omitted]